# Atomically dispersed Pb ionic sites in PbCdSe quantum dot gels enhance room-temperature NO$_2$ sensing

Xin Geng[1,9], Shuwei Li[2,3,4,9], Lalani Mawella-Vithanage[1], Tao Ma[5], Mohamed Kilani [6], Bingwen Wang[7], Lu Ma[8], Chathuranga C. Hewa-Rahinduwage[1], Alina Shafikova[1], Eranda Nikolla [7], Guangzhao Mao[6], Stephanie L. Brock [1✉], Liang Zhang [2,3,4✉] & Long Luo [1✉]

Atmospheric NO$_2$ is of great concern due to its adverse effects on human health and the environment, motivating research on NO$_2$ detection and remediation. Existing low-cost room-temperature NO$_2$ sensors often suffer from low sensitivity at the ppb level or long recovery times, reflecting the trade-off between sensor response and recovery time. Here, we report an atomically dispersed metal ion strategy to address it. We discover that bimetallic PbCdSe quantum dot (QD) gels containing atomically dispersed Pb ionic sites achieve the optimal combination of strong sensor response and fast recovery, leading to a high-performance room-temperature p-type semiconductor NO$_2$ sensor as characterized by a combination of ultra–low limit of detection, high sensitivity and stability, fast response and recovery. With the help of theoretical calculations, we reveal the high performance of the PbCdSe QD gel arises from the unique tuning effects of Pb ionic sites on NO$_2$ binding at their neighboring Cd sites.

[1] Department of Chemistry, Wayne State University, Detroit, MI, USA. [2] Center for Combustion Energy, Tsinghua University, Beijing, China. [3] School of Vehicle and Mobility, Tsinghua University, Beijing, China. [4] State Key Laboratory of Automotive Safety and Energy, Beijing, China. [5] Michigan Center for Materials Characterization, University of Michigan, Ann Arbor, MI, USA. [6] School of Chemical Engineering, University of New South Wales, Sydney, NSW, Australia. [7] Department of Chemical Engineering and Material Science, Wayne State University, Detroit, MI, USA. [8] National Synchrotron Light Source II, Brookhaven National Laboratory, Upton, NY, USA. [9] These authors contributed equally: Xin Geng, Shuwei Li. ✉email: sbrock@chem.wayne.edu; zhangbright@tsinghua.edu.cn; long.luo@wayne.edu

Nitrogen dioxide ($NO_2$) in the atmosphere is of great concern due to its adverse effects on human health and the environment[1–4]. Short-term exposure to a high concentration of $NO_2$ irritates the human respiratory system, causing respiratory distress symptoms such as coughing, wheezing, and difficulty breathing. Long-term exposure to even tens of ppb-level $NO_2$ can cause asthma, particularly for children and the elderly[5]. Most recently, the $NO_2$ concentration in the atmosphere was also found to be positively associated with both COVID-19 case-fatality rate and the overall mortality rate, which show an increase of 11.3% and 16.2% per interquartile range increase in $NO_2$ (4.6 ppb), respectively[6]. $NO_2$ also contributes to environmental problems, such as acid rain, atmospheric haze, nutrient pollution, etc[4]. As a result, the U.S. Environmental Protection Agency (EPA) and European Environmental Agency (EEA) have specified the maximum annual average concentration of $NO_2$ in outdoor air to be 53 (ref. [7]) and 40 ppb[8], respectively. Detection of ppb-level $NO_2$ is typically achieved by chemiluminescence[9], but chemiluminescence analyzers are expensive and inconvenient for real-time and in-field measurements. Despite extensive research efforts focused on developing low-cost alternatives[10–22], the existing low-cost commercial $NO_2$ sensors still cannot provide reliable ppb-level detection in a real-world setting[23–25]. Therefore, there is a need for inexpensive $NO_2$ sensors capable of rapid and reliable sensitivity in the ppb range.

Metal chalcogenide quantum dot (QD) gels have recently emerged as a group of promising materials for gas sensing because of their small crystallite size (high surface-to-volume ratio), three-dimensional (3-D) mesoporous structure (fast gas diffusion), connected network (facilitated electronic communication), and rich chemistry (easy surface modification)[26,27]. Moreover, the process of gelation has the added benefit of partially or entirely stripping organic ligands from the particle surface, thereby substantially increasing the number of active surface sites for gas molecules to bind, improving the sensing performance. Recently, we reported the high $NO_2$ sensing performance of a CdS QD gel at room temperature, which demonstrated high selectivity, an ultra-low (measured, not extrapolated) limit of detection (LOD = 11 ppb), a short response time ($t_{res}$ = ~29 s) and recovery time ($t_{rec}$ = ~28 s)[26]. There was, however, one critical limitation for the CdS QD gel sensor: a relatively low response (0.009% per ppb $NO_2$, red square in Fig. 1a). The analysis of 110 state-of-the-art p-type semiconductor $NO_2$ room-temperature gas sensors in the literature (gray dots in Fig. 1a) reveals such trade-off between sensor response and recovery time is very common because a low sensor response is often correlated with weak adsorption of gas analytes on the sensor surface, facilitating the desorption and sensor recovery. Replacing S in the CdS QD gel with Se did not substantially change the performance (purple pentagon in Fig. 1a), possibly because of the chemical and structural similarity between CdS and CdSe. In contrast, replacing Cd with Pb led to a record-high response: 0.08%/ppb for PbS (blue triangle, Fig. 1a) and 0.075%/ppb for PbSe (yellow diamond, Fig. 1a) and low measured LOD (3 ppb), but they suffer from long $t_{rec}$ (~300 s for PbS and ~240 s for PbSe). An ideal gas sensor should combine high response and short recovery time (purple star in Fig. 1a).

Maximizing response while minimizing recovery time requires fine-tuning of the $NO_2$ binding on the sensing surface. One possible strategy is to use two different metals (i.e., form bimetallic materials) to generate sites with different binding energetics from the monometallic surfaces. This strategy has been widely adopted in the field of catalysis, which also requires an optimal binding of adsorbates to achieve high catalytic performance[28–37]. Partial cation exchange is a facile and powerful synthetic method for preparing bimetallic chalcogenide nanoparticles[38–47]. During partial cation exchange reactions, only a fraction of the cations in the nanoparticles are replaced by cations in the solution phase. The extent of partial cation exchange can be precisely controlled by the cation concentration and reaction time. In principle, partial cation exchange reactions of metal chalcogenide QD gels are also feasible but have not been extensively studied[48–51].

Here, we show the bimetallic $Pb_xCd_{1-x}Se$ QD gels with only atomically dispersed Pb sites result in the ideal combination of high response and short recovery time, leading to a high-performance room-temperature p-type semiconductor $NO_2$ gas sensor with a combination of ultra-low LOD (3 ppb), high sensitivity (0.06%/ppb), short $t_{res}$ (~28 s), and $t_{rec}$ (~60 s). Density functional theory (DFT) calculations suggest that the high performance of the $Pb_xCd_{1-x}Se$ QD gel is caused by the unique tuning effects of atomically dispersed Pb ionic sites on $NO_2$ binding at their neighboring Cd sites.

## Results

**Synthesis of $Pb_xCd_{1-x}Se$ QD gels.** Based on prior work demonstrating (1) slow gelation kinetics for cubic polymorphs (i.e., PbSe) relative to hexagonal polymorphs, and (2) facilitated cation exchange on ligand-stripped surfaces[52,53], our strategy for targeting $Pb_xCd_{1-x}Se$ QD gels involves initial synthesis of hexagonal (wurtzite) CdSe, subsequent gelation (induced by ligand stripping), and ultimately ion-exchange of $Cd^{2+}$ for $Pb^{2+}$as illustrated in Fig. 1b. First, nearly monodisperse thiolate-capped CdSe QDs with sizes of 3.0 ± 0.3 nm were prepared according to a modified hot-injection method followed by ligand exchange (Fig. 1c, d and Supplementary Fig. 1a)[54]. Next, CdSe QDs were crosslinked by electro-oxidative removal of the protecting thiolate ligands (as disulfides) in conjunction with the electro-oxidative formation of di-selenide linkages between CdSe QDs to form a macroscopic 3-D connected pore-matter CdSe QD gel[26]. The synthesized CdSe QD gel exhibits a mesoporous network comprising CdSe building blocks with the same size (3.0 ± 0.4 nm) as the starting QDs (Fig. 1e, f and Supplementary Fig. 1b). The lattice fringes of the CdSe QD gel can be assigned to different planes of hexagonal CdSe (Fig. 1f), indicating that the gel is polycrystalline and its QD building blocks are randomly oriented. Finally, $Pb_xCd_{1-x}Se$ QD gels were synthesized via a cation exchange process wherein $x$ is controlled by the concentration of $Pb(NO_3)_2$ in the exchange solution[48,51]: a greater $Pb(NO_3)_2$ concentration leads to more Pb incorporation in the $Pb_xCd_{1-x}Se$ QD gels. The experimentally measured Pb concentration was determined by X-ray photoelectron spectroscopy (XPS) and inductively coupled plasma mass spectrometry (ICP-MS) (Supplementary Table 1). As $x$ increases from 0 to 1.0, the color of the $Pb_xCd_{1-x}Se$ QD gels gradually changed from orange to black (Supplementary Fig. 2).

**Structural characterization and modeling of $Pb_xCd_{1-x}Se$ QD gels.** Scanning transmission electron microscopic (STEM) images of $Pb_xCd_{1-x}Se$ QD gels with $x \leq 0.17$ show similar crystallite sizes (~3 nm) as the original CdSe QD gel (Fig. 1g, h and Supplementary Fig. 1c–f). However, as Pb content increases beyond this point, the crystallite size in the gel increases, reaching 8.4 ± 1.1 nm for complete exchange ($x = 1$, Supplementary Fig. 1g and Supplementary Fig. 3). This ripening may be due to structural disruption from the rapid cation exchange between $Cd^{2+}$ and $Pb^{2+}$ under forcing conditions created by high $Pb^{2+}$ concentrations. This can potentially be remedied by slowing the kinetics, e.g., by adjusting the solvent (playing off differences in solvation energy for $Cd^{2+}$ and $Pb^{2+}$) and/or using a lower ionic concentration of the exchanging ion. For the present study, we selected to focus on $Pb_xCd_{1-x}Se$ QD gels with $x \leq 0.40$ to

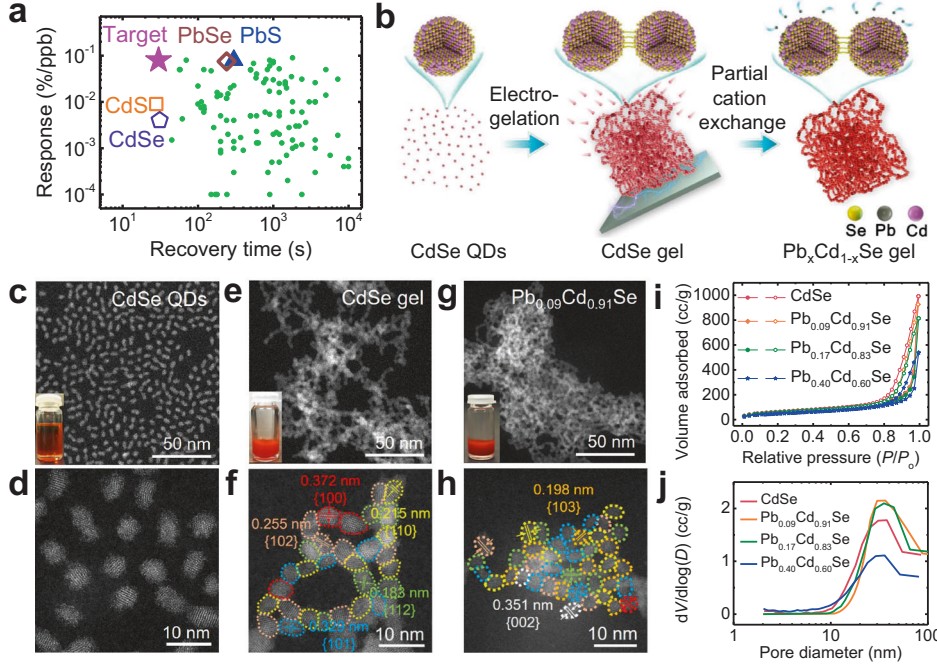

**Fig. 1 Synthesis of $Pb_xCd_{1-x}$Se QD gels. a** Comparison of CdS, CdSe, PbS, and PbSe QD gels with 110 state-of-the-art room-temperature p-type semiconductor $NO_2$ sensors in the literature in terms of sensor response and recovery time. **b** Schematic diagram for the synthesis of $Pb_xCd_{1-x}$Se QD gels ($x = 0.003, 0.02, 0.04, 0.09, 0.17, 0.40$, and $1.0$) via cation exchange using different $Pb(NO_3)_2$ concentrations. **c–h** STEM images of **c, d** CdSe QDs, **e, f** CdSe gel, and **g, h** $Pb_{0.09}Cd_{0.91}$Se gel at low and high magnification, inset: the corresponding photographs. Crystallites in the high-resolution images are color-coded based on their lattice fringes, corresponding to the {100}, {101}, {102}, {103}, {110}, {112}, and {002} planes of hexagonal CdSe. **i** Nitrogen adsorption–desorption isotherms and **j** Barrett–Joyner–Halenda pore-size distributions of CdSe, $Pb_{0.09}Cd_{0.91}$Se, $Pb_{0.17}Cd_{0.83}$Se, and $Pb_{0.40}Cd_{0.60}$Se QD gels.

minimize potential contributions from morphological changes on the observed sensing performance.

The porosity of $Pb_xCd_{1-x}$Se QD gels was analyzed by nitrogen physisorption after supercritical drying to produce aerogels (Supplementary Fig. 4), which produced type-IV isotherms, characteristic of a mesoporous material (Fig. 1i). The surface areas of $Pb_xCd_{1-x}$Se QD aerogels are similar to that of the native CdSe QD aerogel based on the Brunauer–Emmett–Teller (BET) model[55] (177–205 $m^2$/g for $Pb_xCd_{1-x}$Se vs 209 $m^2$/g for CdSe in Supplementary Table 2). Figure 1j illustrates the pore-size distributions for $Pb_xCd_{1-x}$Se and CdSe QD aerogels, obtained by fitting the desorption branch of the isotherms to the Barrett–Joyner–Halenda model[56]. The average pore diameters and cumulative pore volumes for $Pb_xCd_{1-x}$Se QD aerogels were calculated to be 17.9–22.5 nm and 0.8–1.5 $cm^3$/g, respectively, similar to values for the CdSe QD aerogels (22.3 nm and 1.3 $cm^3$/g, respectively).

The atomic structure of the $Pb_xCd_{1-x}$Se QD gels was further characterized by high-angle annular dark-field scanning transmission electron microscopy (HAADF-STEM), energy-dispersive X-ray spectroscopy (EDS) elemental mapping, powder X-ray diffraction (PXRD), XPS, and X-ray absorption spectroscopy (XAS). For all $Pb_xCd_{1-x}$Se QD gels, the HAADF-STEM images show atomic sites with a higher contrast (marked by yellow circles in Fig. 2a–d, left), which could be assigned to atomically dispersed Pb cation sites due to the Z-contrast in HAADF-STEM. The contrast of Pb ($Z = 82$) vs Cd ($Z = 48$) cations is clearly seen in the intensity profiles (Fig. 2e) integrated from the marked area 1 and 2 of $Pb_{0.17}Cd_{0.83}$Se QD gel (Fig. 2c). Moreover, the number density of Pb ionic sites increase with the increasing Pb content in the gel. Note that Pb is only seen as atomically dispersed ionic sites in the gels with low Pb contents, such as $x = 0.04$ and 0.09. For

$Pb_xCd_{1-x}$Se QD gels with $x \geq 0.17$, nanometer-sized high-contrast regions were also observed in HAADF-STEM (Fig. 2f and Supplementary Fig. 5f). The fast Fourier transform (FFT) patterns from these regions were indexed to the cubic PbSe phase. The EDS mapping results also confirmed the structural transition from a uniform dispersion of Pb atomic sites on the CdSe surface at low Pb contents (Fig. 2a, b, right and Supplementary Fig. 6a, b) to the phase separation of PbSe and CdSe at high Pb contents (Fig. 2c, d, right, and Supplementary Fig. 6c, d).

The PXRD patterns of $Pb_xCd_{1-x}$Se QD gels in Fig. 3a are consistent with the microscopy data, showing phase segregation of PbSe with increasing Pb content. At $x \leq 0.17$, only the characteristic peaks of hexagonal CdSe (PDF 00-008-0459, wurtzite) were present, whereas for $x > 0.17$, the characteristic peaks of cubic PbSe (PDF 00-006-0354, rock salt) appear. Note that even though the cubic phase PbSe was observed in the STEM images of $Pb_{0.17}Cd_{0.83}$Se QD gel, its PXRD peaks were not found, likely because their signals were below the detection limit of the XRD instrument.

XPS measurements were next performed to examine the chemical states of Pb4$f_{5/2}$ and Cd3$d$ in $Pb_xCd_{1-x}$Se QD gels during the composition-dependent structural transition. The Cd3$d$ peaks did not show any notable differences for all $Pb_xCd_{1-x}$Se and CdSe QD gels (Fig. 3b). In contrast, the Pb4$f_{5/2}$ peak of the $Pb_xCd_{1-x}$Se QD gels upshifted from 143.5 eV at $x < 0.17$ to 143.8 eV at $x \geq 0.17$ as a result of the change in the chemical environment of Pb from atomically dispersed Pb ionic sites in a hexagonal CdSe matrix to Pb in the cubic phase PbSe (Supplementary Fig. 7 and Supplementary Table 3)[57]. Similar XPS peak shifts have been previously reported for other single-atom materials[58–60]. For example, Wu et al.[58] found that the Au4$f$ peaks of Au single atoms

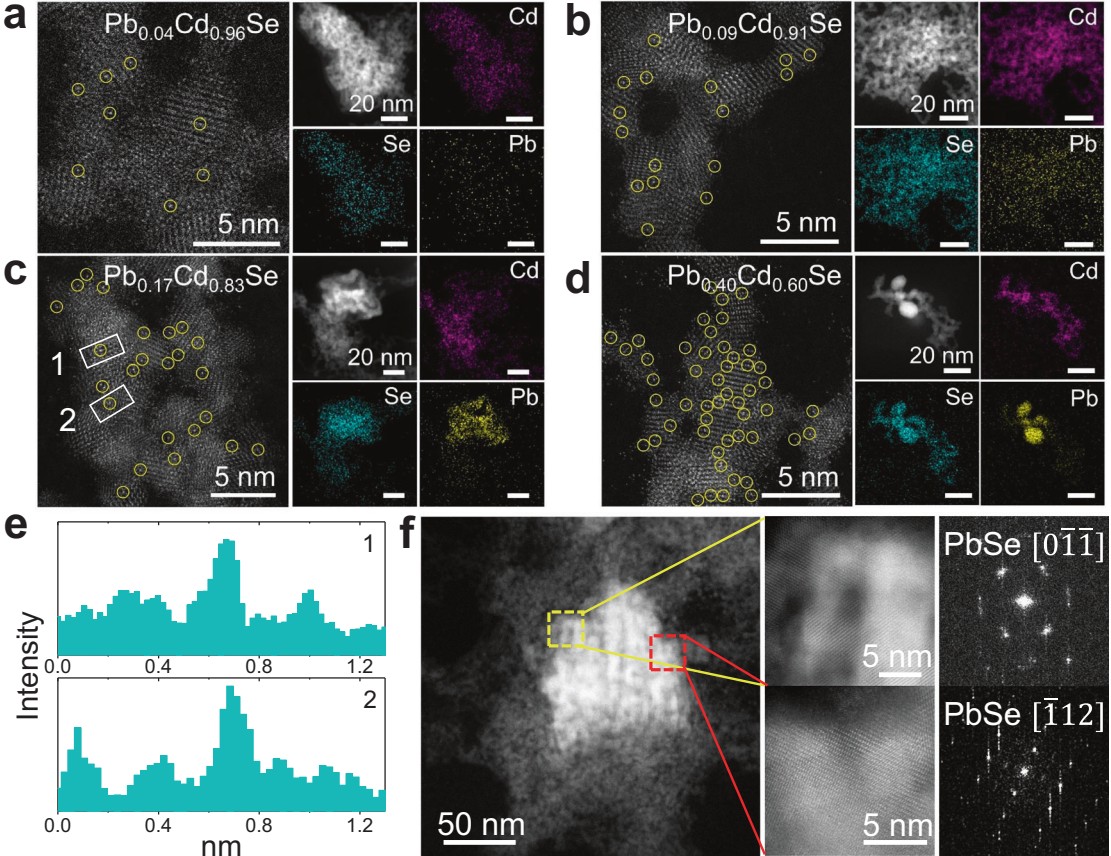

**Fig. 2 HAADF-STEM characterization and EDS mapping of $Pb_xCd_{1-x}Se$ QD gels. a–d** HAADF-STEM images and EDS elemental maps of **a** $Pb_{0.04}Cd_{0.96}Se$, **b** $Pb_{0.09}Cd_{0.91}Se$, **c** $Pb_{0.17}Cd_{0.83}Se$, and **d** $Pb_{0.40}Cd_{0.60}Se$ QD gels. **e** Intensity profiles of $Pb_{0.17}Cd_{0.83}Se$ gel in the marked area 1 and 2 in panel **c**. **f** Low- and high-magnification HAADF-STEM images of a high-contrast region of $Pb_{0.17}Cd_{0.83}Se$ QD gel and the corresponding FFT images from the marked regions.

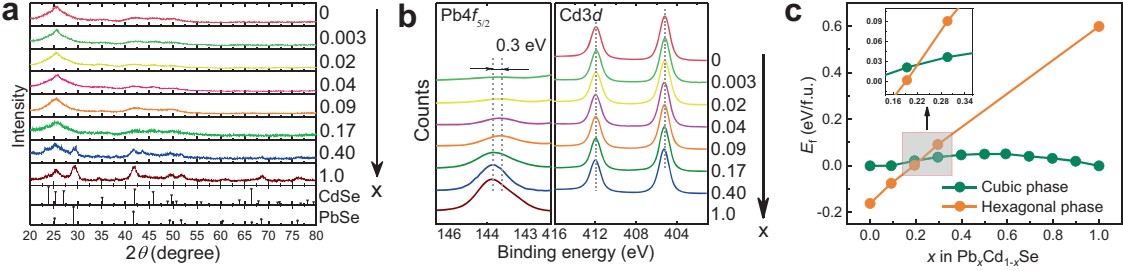

**Fig. 3 Structural characterization of $Pb_xCd_{1-x}Se$ QD gels and related machine learning (ML) simulation. a** PXRD patterns of $Pb_xCd_{1-x}Se$ QD gels with $x = 0.003, 0.02, 0.04, 0.09, 0.17, 0.40$, and $1.0$. The stick diagrams show the PXRD patterns of hexagonal CdSe and cubic PbSe as references. **b** XPS results of the Cd$3d$ and Pb$4f_{5/2}$ regions for the QD gels. **c** Calculated thermodynamic convex hull diagram of $Pb_xCd_{1-x}Se$ in cubic and hexagonal phases with respect to cubic PbSe and cubic CdSe using the converged ML surrogate model (Supplementary Fig. 10). Note that the $Pb_xCd_{1-x}Se$ hexagonal supercells at $x > 0.3$ were unstable with severe deformation during DFT calculations; therefore, we only considered the $Pb_xCd_{1-x}Se$ hexagonal supercells with $x = 0, 0.1, 0.2, 0.3$, or $1.0$. The inset shows the expanded view of the crossover region.

on a CuO support shifted downward by 0.3 eV relative to those of bulk Au.

XAS measurements were also carried out to further investigate the chemical states and electronic structures of Cd and Pb atoms in the $Pb_xCd_{1-x}Se$ QD gels. As shown in Supplementary Fig. 8, X-ray absorption near edge structure (XANES) of the Cd K-edge for the $Pb_xCd_{1-x}Se$ and CdSe QD gels did not show any obvious differences in absorption energy. The first derivative peak positions are CdSe: 26716.7 eV, $Pb_{0.09}Cd_{0.91}Se$: 26716.4 eV, and $Pb_{0.4}Cd_{0.6}Se$: 26717.6 eV (note: the step size of 1 eV used in the XANES measurements), in good agreement with the XPS results.

The prominent scattering pathway at 2.2 Å in the extended X-ray absorption fine structure (EXAFS) spectra of the Cd K-edge is attributed to the interatomic scattering pathway of Cd–Se (Supplementary Fig. 8). The EXAFS data fitting results in Supplementary Table 4 show a Cd–Se distance of 2.571–2.606 Å with a coordination number of 2.8–3.8 for these three samples (bulk = 4), which is close to the theoretical coordination number for ~3 nm CdSe nanoparticles (the coordination number expected for a 25-Å spherical particle is 2.9–3.8, depending on the exact size and origin of the sphere with respect to the lattice)[61]. These results suggest a similar chemical environment for Cd in $Pb_xCd_{1-x}Se$

QD gels. However, because of the X-ray absorption interference between the Se K-edge (12657.8 eV) and the Pb L-edge (13035.2 eV), the XAS signal of the Pb $L_{III}$ edge was too noisy at the edge region for EXAFS analysis to obtain the coordination environment of Pb in the gel samples (Supplementary Fig. 9a, d). The successive absorption signals at Pb $L_{II}$ and $L_I$ edges were also too weak to analyze with confidence (Supplementary Fig. 9b–f).

Taking all the experimental evidence together, we conclude that Pb in $Pb_xCd_{1-x}Se$ QD gels can exist in two forms: atomically dispersed Pb ionic sites in the hexagonal CdSe matrix and Pb in the cubic phase PbSe. At $x < 0.17$, Pb ions are only present as atomically dispersed ionic sites, while at $x \geq 0.17$, both forms exist.

To understand the structural transition of $Pb_xCd_{1-x}Se$ QD gels as a function of Pb content, we performed theoretical calculations to construct the thermodynamic convex hull for $Pb_xCd_{1-x}Se$. In the calculations, we employed an active machine learning (ML) scheme combined with DFT calculations (Supplementary Figs. 10–15) to explore a much larger $Pb_xCd_{1-x}Se$ configuration space than the conventional DFT approach, which typically considers only a few representative structures due to the high computational cost[62,63]. Figure 3c shows the thermodynamic convex hull for cubic (green dots) and hexagonal (orange dots) $Pb_xCd_{1-x}Se$, where the lowest formation energies among over 50,000 explored configurations at different Pb contents are plotted. The energy of $Pb_xCd_{1-x}Se$ ($E_f$) is referenced to that of cubic PbSe and CdSe ($E_{PbSe}$ and $E_{CdSe}$), $E_f = E - xE_{PbSe} - (1-x)E_{CdSe}$. The converged surrogate model has a prediction error < 0.002 eV/f.u. when benchmarked with DFT calculations. As expected, pristine hexagonal CdSe is thermodynamically more stable than its cubic counterpart[64]. However, the increasing Pb content in the $Pb_xCd_{1-x}Se$ significantly destabilizes the hexagonal phase, reducing the energy difference between the hexagonal phase and cubic phase. The crossover of the two curves in Fig. 3c suggests the structural transition from hexagonal phase to cubic phase occurs at $x = \sim0.21$, which is in good agreement with our experimental observation.

**NO₂ gas-sensing performance.** $Pb_xCd_{1-x}Se$ gel sensors were prepared by drop-casting 10 μL of wet gel (or alcogel as the solvent is methanol) onto a sensor substrate patterned with an interdigitated electrode, followed by drying in air. For regular sensing tests, commercial alumina substrates were used because of their cost-effectiveness. For the homebuilt wireless portable-sensing device tests, silicon-based substrates were fabricated using photolithography to be compatible with the device (Supplementary Figs. 16 and 17). The dried xerogel film exhibits a highly porous surface morphology (Supplementary Fig. 18) and a thickness of ~3.1 μm (Supplementary Fig. 19). The gel film thickness can be varied by controlling the amount of wet gel deposited onto the substrate. Although thinner films are expected to afford higher sensing performance, including higher response and faster response and recovery, the reduced film thickness dramatically increases the sensor resistance, imposing a technical challenge in measuring resistance (Supplementary Fig. 19 and Supplementary Table 5). As a result, we performed all the sensing tests at the film thickness of 3.1 μm. The mesoporous structure of gel was still retained in the xerogel film, although the surface area decreased significantly compared to the aerogel due to partial pore collapse during ambient drying (Supplementary Fig. 20 and Supplementary Table 2). The sensing tests were carried out at room temperature using a homebuilt apparatus[26]. The sensor response is defined as $|R_a-R_g|/R_a$, where $R_a$ and $R_g$ are the resistance in the presence of air and target gas, respectively. $t_{res}$ and $t_{rec}$ represent the time required when the resistance changes

90% at the exposure and removal stage of the target gas, respectively.

Figure 4a shows the responses of $Pb_xCd_{1-x}Se$ QD gels ($x = 0$, 0.003, 0.02, 0.04, 0.09, 0.17, 0.40, and 1.0) to various NO₂ concentrations from 3 to 1.32 ppm in air at room temperature. All $Pb_xCd_{1-x}Se$ and CdSe QD gels exhibit the characteristic behavior of p-type semiconductors, whose resistance decreases upon exposure to NO₂ and returns to the initial resistance after the removal of NO₂. The contact resistance between QD gel film and electrodes is ~2 orders of magnitude smaller than the gel film's resistance and, thus, is negligible (Supplementary Fig. 21). The sensor response increases linearly with increasing NO₂ concentration for all gel sensors (Fig. 4b). However, the presence of Pb in the gel dramatically improves the sensor response by over 1600% (from 0.004%/ppb for CdSe to 0.065%/ppb for $Pb_{0.4}Cd_{0.6}Se$, Fig. 4b). The high sensor response has led to a LOD of 3 ppb for $Pb_xCd_{1-x}Se$ QD gels with $x \geq 0.09$ (Supplementary Fig. 22), which more than meets the LOD requirements by EPA (53 ppb)[7] and EEA (40 ppb)[8] for NO₂ sensing. Note that the LODs reported here are all experimentally measured LODs, not calculated LODs using the $3\sigma$ rule. More interestingly, the increase in sensor response is not a linear function of the Pb content. For example, the sensor response to 1.32 ppm NO₂ has already increased by 1300% as $x$ is slightly changed from 0 to 0.09, whereas it only increases by another 486% when $x$ is further increased to 1.0 (Fig. 4c and Supplementary Table 6). In contrast, $t_{rec}$ increases nearly linearly with increasing Pb content and the change in $t_{res}$ is negligible as $x$ is varied from 0 to 1.0 (Fig. 4c and Supplementary Table 6). The different dependences of sensor response and $t_{rec}$ on the Pb content in $Pb_xCd_{1-x}Se$ QD gels have led to an optimal combination of high sensor response (0.06%/ppb) and short $t_{rec}$ (~60 s) in $Pb_{0.09}Cd_{0.91}Se$ QD gel sensor. Compared to 110 state-of-the-art room-temperature NO₂ gas sensors based on p-type semiconductors in the literature (green lines in Fig. 4d), the $Pb_{0.09}Cd_{0.91}Se$ gel sensor is superior, demonstrating a combination of high sensor response, low LOD, and short $t_{res}$ and $t_{rec}$ (Fig. 4d and Supplementary Table 7). Note that the loading of $Pb(NO_3)_2$ in a CdSe QD gel matrix without cation exchange (achieved by mixing $Pb(NO_3)_2$ with CdSe QD gel in hexane) does not lead to any sensing performance improvement (Supplementary Fig. 23), suggesting the high performance of $Pb_{0.09}Cd_{0.91}Se$ QD gel is a direct result of cation exchange.

Cycling stability and analyte selectivity are essential for the design of practical sensors. We tested the stability of the $Pb_{0.09}Cd_{0.91}Se$ gel sensor by exposing it to 440 ppb NO₂ for 560 consecutive exposure/removal cycles. During this 75-h-long stability test, the sensor response only varied by ~10%, and $t_{res}$ and $t_{rec}$ stayed constant at $31 \pm 3$ and $63 \pm 10$ s (Fig. 4e and Supplementary Fig. 24), suggesting the strong gel adhesion to the substrate and high stability of the gel sensors. The $Pb_{0.09}Cd_{0.91}Se$ gel sensor also exhibited excellent selectivity toward NO₂ with at least 3.5 times higher response than sulfur dioxide (electron-withdrawing analyte) and ammonia, hydrogen, carbon monoxide, methanol, ethanol, acetone, and formaldehyde (electron-donating analytes) even when the concentrations of the electron-donating molecules are 75-fold that of NO₂ (Fig. 4f).

To further demonstrate the potential real-world applications of the $Pb_{0.09}Cd_{0.91}Se$ QD gel, we built a wireless portable NO₂-sensing device using the QD gel (Supplementary Fig. 25) and compared it with a commercial NO₂ detector (Manufacturer: Forensics, Part Number: 6S-Z1JF-MOYC) purchased from Amazon. The two devices showed similar $t_{res}$ and $t_{rec}$ and provided identical results in the NO₂ concentration range of 100–600 ppb, but the commercial devices failed to detect NO₂ concentrations lower than 100 ppb, whereas our QD gel sensor responded to NO₂ concentrations as low as 10 ppb (Fig. 4g–i,

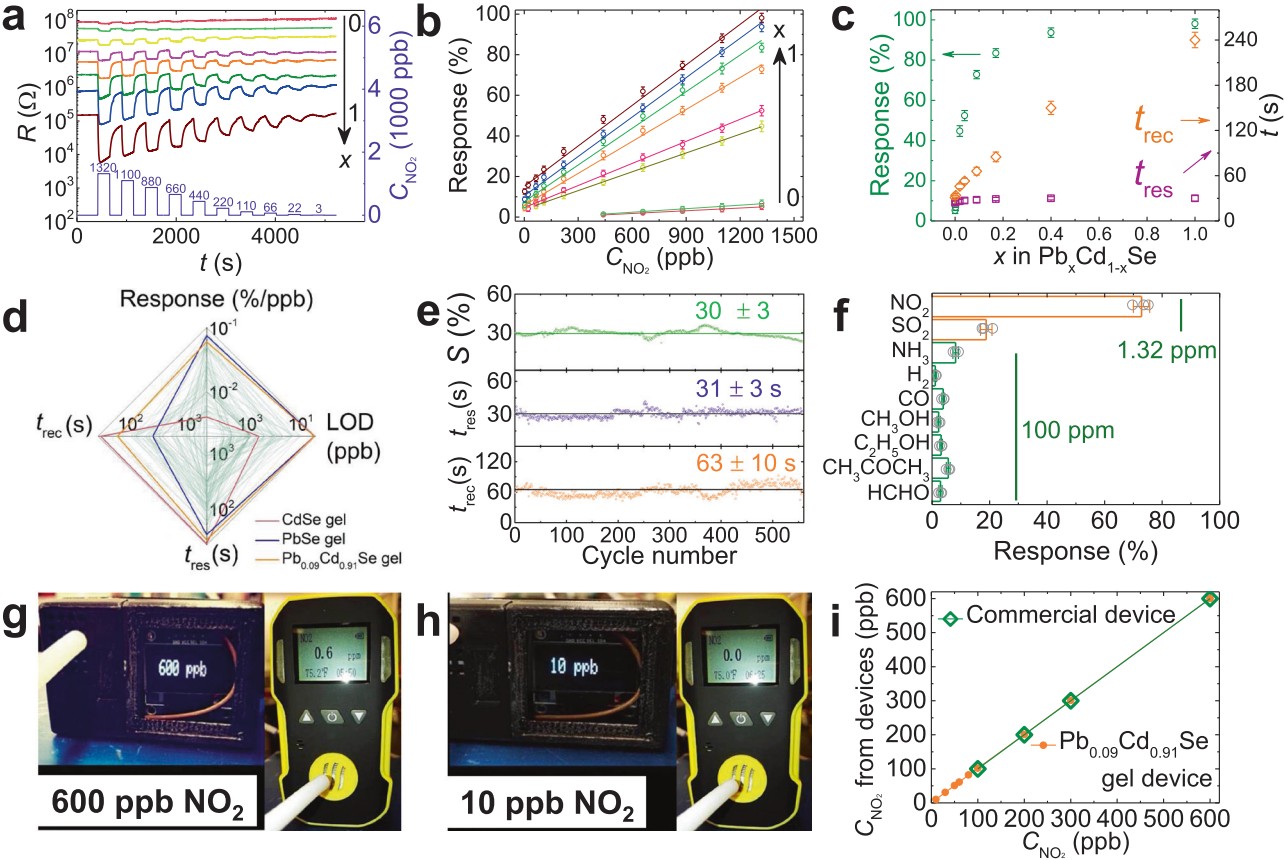

**Fig. 4 Gas-sensing performance of $Pb_xCd_{1-x}Se$ QD gels at room temperature. a** Response–recovery curves of $Pb_xCd_{1-x}Se$ QD gel sensors to $NO_2$ at different concentrations (3 ppb–1.32 ppm) at room temperature ($x = 0.003, 0.02, 0.04, 0.09, 0.17, 0.40,$ and $1.0$). **b** Sensor response vs $NO_2$ concentration ($C_{NO_2}$) plots. **c** Sensor response ($S$), response time ($t_{res}$), and recovery time ($t_{rec}$) of $Pb_xCd_{1-x}Se$ QD gels towards 1.32 ppm $NO_2$ as a function of $x$. **d** Comparison between CdSe, PbSe, $Pb_{0.09}Cd_{0.91}Se$ QD gels, and 110 state-of-the-art room-temperature p-type $NO_2$ gas sensors in the literature. **e** Stability of a $Pb_{0.09}Cd_{0.91}Se$ QD gel sensor during a 75-h-long 560 $NO_2$ exposure/removal cycles. **f** Responses of a $Pb_{0.09}Cd_{0.91}Se$ QD gel sensor to different 100 ppm gases or vapors at room temperature ($NO_2$ and $SO_2$ concentrations are 1.32 ppm). **g, h** Photographs of our homebuilt $Pb_{0.09}Cd_{0.91}Se$ QD gel detector (left) and a commercial $NO_2$ detector (right) bought from Amazon (Manufacturer: Forensics, Part Number: 6S-Z1JF-MOYC) in response to **g** 600 ppb and **h** 10 ppb $NO_2$ at room temperature. **i** Readouts of the $Pb_{0.09}Cd_{0.91}Se$ QD gel sensor and the commercial $NO_2$ detector at different $NO_2$ concentrations (10–600 ppb). The error bars in panels **b**, **c**, **f** are the standard deviations of three independent measurements.

Supplementary Fig. 26, and Supplementary Movie 1). It is worth noting that the Pb and Cd contents by weight in this homebuilt $NO_2$-sensing device are 0.1 and 0.5 ppm, respectively; both are several orders of magnitude lower than the International Standards for Pb (0.1% or 1000 ppm) and Cd (0.01% or 100 ppm) in electrical and electronic devices set by the RoHS (restriction of the use of certain hazardous substances)[65].

**$NO_2$ gas-sensing mechanism.** According to the structural characterization results in Figs. 2 and 3, Pb in the $Pb_{0.09}Cd_{0.91}Se$ QD gel is present as atomically dispersed Pb ionic sites. To understand the contribution of these ionic Pb sites to the superior $NO_2$-sensing performance, we carried out DFT calculations of the $NO_2$ adsorption energy and the degree of charge transfer on various surface structures of $Pb_xCd_{1-x}Se$ (Fig. 5a, b and Supplementary Fig. 27), including a hexagonal CdSe (100) surface (A), a hexagonal CdSe (100) surface covered by a monolayer of PbSe (B), a cubic PbSe (100) surface (C), and a hexagonal CdSe (100) surface with atomically dispersed Pb (D1, D2, E1, E2) or two neighboring Pb (F1, F2). The adsorption energy describes how strongly a $NO_2$ molecule binds to a surface; the stronger the adsorption energy, the longer the recovery time. The charge transfer between the adsorbed $NO_2$ and the semiconducting gel sensor dictates the change in electrical resistivity of the

semiconductor sensor (or the sensor response)[66,67]. Thus, to achieve a combination of strong response and short recovery time, a large degree of electron transfer coupled with weak adsorption energy is desired; however, in practice, charge transfer is generally positively correlated with the adsorption energy.

The DFT results in Fig. 5a for various surface structures of $Pb_xCd_{1-x}Se$ confirm the experimental observation that a strong sensor response is often accompanied by a long recovery time. However, a closer look at these data points shows that, among all surface structures, PbSe surfaces (B and C) have the largest charge transfer and highest adsorption energies, while the hexagonal CdSe surface (A) is the opposite, consistent with the sensing performance of the monometallic PbSe and CdSe QD gels in Fig. 1a. For all bimetallic surfaces, their Cd sites adjacent to Pb show stronger adsorption energies and larger electron transfer than the Cd site on the monometallic CdSe surface (D2, E2, F2 vs A), whereas their Pb sites show weaker adsorption energies and smaller electron transfer than the Pb site on the monometallic PbSe surface (D1, E1, F1 vs B, C). This finding indicates electronic communication between the Pb and Cd sites in these bimetallic surfaces, which has affected $NO_2$ binding and consequently charge transfer on these sites. Interestingly, for bimetallic surfaces with atomically dispersed Pb sites, Cd cations next to a Pb site (D2 and E2) have significantly larger charge

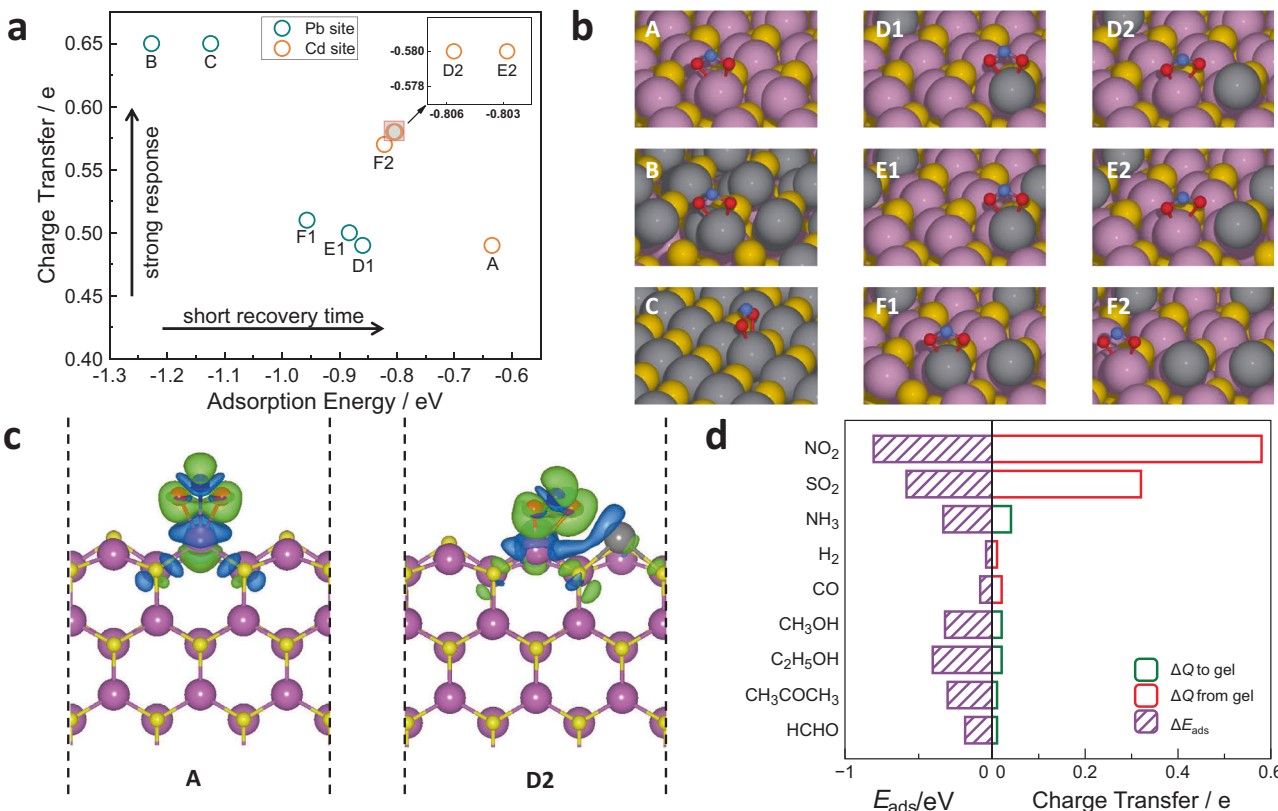

**Fig. 5 DFT calculation results of NO$_2$ adsorption energies and the degrees of charge transfer to understand the sensing performances of the Pb$_x$Cd$_{1-x}$Se gel sensor. a** Plot of charge transfer vs adsorption energy for NO$_2$ adsorption on various surface structures of Pb$_x$Cd$_{1-x}$Se: hexagonal CdSe (100) surface (A), a hexagonal CdSe (100) surface covered by a monolayer of PbSe (B), cubic PbSe (100) surface (C), and a hexagonal CdSe (100) surface with atomically dispersed Pb (D1, D2, E1, E2) or two neighboring Pb (F1, F2). **b** The corresponding NO$_2$ adsorption geometries. Pb, Cd, Se, N, and O atoms are shown as gray, purple, yellow, blue, and red balls, respectively; their top and side view figures are provided in Supplementary Fig. 27. **c** Calculated differential valence-electron charge densities of NO$_2$ adsorption for A and D2 in panel **b** ($\triangle\rho = \rho_{NO_2^*} - \rho_* - \rho_{NO_2}$); charge depletion and accumulation are illustrated by blue and green, respectively. **d** Calculated adsorption energies (left side) and charge transfer (right side) for different gases on the D2 surface in panel **b**. The charge transferred from and to the surface is plotted in red and green, respectively.

transfer with the adsorbate than the Pb neighboring sites (D1 and E1) but share comparable NO$_2$ adsorption energies. This suggests that NO$_2$ can bind either at the Pb or Cd sites on these surfaces, but the sensor response is dominated by NO$_2$ binding at the Cd sites adjacent to Pb due to the more significant charge transfer. In contrast, for bimetallic surfaces where two Pb sites are adjacent to each other, the situation is quite different. Even though the Cd sites (F2) still have significantly larger charge transfer than the Pb sites (F1), the NO$_2$ adsorption energies become notably stronger at the Pb sites than the Cd sites, causing preferential NO$_2$ adsorption at the Pb sites over the Cd ones. Thus, the sensor response is dominated by NO$_2$ binding at the Pb sites with less charge transfer than the Cd sites.

The above findings show that (1) the Cd sites adjacent to Pb cations in Pb$_x$Cd$_{1-x}$Se surface lead to the "desired" sites for NO$_2$ adsorption since they provide the largest electronic communication with NO$_2$ under similar adsorption strength to that of Pb sites; and (2) the key function of the atomically dispersed Pb sites is to reduce the competition for NO$_2$ binding between the Cd and Pb sites, allowing the Cd sites to be "functional". We further analyzed the differential electron density of NO$_2$ adsorption on the Cd sites on the Pb$_x$Cd$_{1-x}$Se bimetallic surface to understand the origin of the large charge transfer at these sites. The result in Fig. 5c shows the neighboring Pb site acts as an electron donor, promoting the electron transfer from Pb$_x$Cd$_{1-x}$Se to the adsorbed NO$_2$ at the Cd site.

Additionally, we performed a DFT calculation of the adsorption energies and charge transfer for all nine gases tested in the

experiment to understand the origin of the selectivity of Pb$_{0.09}$Cd$_{0.91}$Se gels toward NO$_2$. We used geometry D2 as the representative structure for Pb$_{0.09}$Cd$_{0.91}$Se in this calculation. As shown in Fig. 5d, the charge transfer values have a similar pattern with the sensor responses in Fig. 4f, suggesting that the good selectivity for NO$_2$ results from the remarkable charge transfer from the surface to NO$_2$ (the optimized structure of the most stable adsorption geometries is provided in Supplementary Fig. 28).

## Discussion

In this work, we synthesized bimetallic Pb$_x$Cd$_{1-x}$Se QD gels ($x = 0$, 0.003, 0.02, 0.04, 0.09, 0.17, and 0.40) via a three-step approach consisting of CdSe QD synthesis, electrogelation, and partial cation exchange. All the Pb$_x$Cd$_{1-x}$Se QD gels showed similar surface morphology, crystallite size, surface area, and porosity as the precursor CdSe QD gel. The HAADF-STEM, EDX mapping, PXRD, and XPS results reveal that the Pb in the Pb$_x$Cd$_{1-x}$Se QD gels can exist in two different forms depending on the Pb content. At low Pb content ($x < 0.17$), Pb sites are atomically dispersed in the gel. When $x$ further increases ($x \geq 0.17$), phase separation of cubic PbSe starts to take place so that atomically dispersed Pb sites and cubic PbSe coexist in the gels up until complete replacement of Cd for Pb ($x = 1$). Theoretically calculated thermodynamic convex hull using an active ML strategy shows that the bimetallic Pb$_x$Cd$_{1-x}$Se hexagonal phase is thermodynamically

stable for a Pb content lower than 20%, in good agreement with experimental results.

The $NO_2$ gas-sensing results show a composition-dependent gas-sensing performance of the $Pb_xCd_{1−x}Se$ QD gels. The optimal combination of strong sensor response and fast recovery is achieved for $x = 0.09$, where Pb is present only as atomically dispersed metal ions. Compared to 110 state-of-the-art room-temperature $NO_2$ gas sensors based on p-type semiconductors in the literature, the $Pb_{0.09}Cd_{0.91}Se$ QD gel sensor demonstrates high performance with the combination of ultra-low LOD (3 ppb), high response (0.06%/ppb), and short $t_{res}$ (~28 s) and $t_{rec}$ (~60 s). The $Pb_{0.09}Cd_{0.91}Se$ gel sensor also exhibits excellent stability concerning the sensor response, $t_{res}$, and $t_{rec}$, which varies by only ~10% during a 75-h-long stability test. A high selectivity toward $NO_2$ vs eight different common gases is also achieved. Furthermore, we fabricated a wireless portable $NO_2$ device using the $Pb_{0.09}Cd_{0.91}Se$ gel and compared it with a commercial $NO_2$ detector in real time. The results reveal that our $NO_2$-sensing device is highly reliable for detecting ppb levels of $NO_2$. Remarkably, compared to the commercial device, our device works well even in the range of 10–100 ppb, fulfilling the LOD requirements of the EPA (53 ppb) and EEA (40 ppb), suggesting great potential for commercial markets.

The DFT calculation results suggest that it is the Cd sites, rather than the Pb sites, on the bimetallic $Pb_xCd_{1−x}Se$ QD gel surface that are the adsorption sites responsible for the exceptional $NO_2$-sensing performance because they provide a significantly larger charge transfer but comparable adsorption energy, relative to the Pb sites, addressing the trade-off between the response and recovery time. The atomically dispersed Pb ionic sites are quite different from Pb sites in the PbSe matrix, serving to transfer electron density to adjacent Cd cations, making them better electron donors to $NO_2$ and enhancing the response. Our findings are significant because trade-offs are common barriers in sensing and catalytic performance, and this work shows that bimetallic structures with atomically dispersed metal ion geometries can be a strategy to achieve an optimal balance between adsorbate binding energy and extent of charge transfer, leading to enhanced functionality.

## Methods

**Chemicals and materials**. Selenium powder (Se, 99.99%), trioctylphosphine oxide (TOPO, 99%), tetradecylphosphonic acid (TDPA), 1- thioglycolic acid (TGA), tetramethylammonium hydroxide (TMAH), tetrabutylammonium hexafluorophosphate ($NBu_4PF_6$, 98%), and Pt foil (0.125–0.135 mm thickness, 99.9%) were purchased from Sigma-Aldrich; cadmium oxide (CdO, 99.99%) and trioctylphosphine (TOP>85%) were purchased from Strem chemicals; lead nitrate (Pb($NO_3$)$_2$, 99.3%) was purchased from Fisher; all chemicals were of analytical grade and used without further purification.

**Synthesis of CdSe QDs**. CdSe QDs were synthesized using a modified hot-injection approach[54]. A mixture of 0.0508 g (0.4 mmol) CdO, 0.16 g TDPA, and 8.0 g TOPO were heated in a 100 mL Schlenk flask in vacuum at 150 °C for 20 min. Next, the vacuum was removed, and a continuous argon flow was introduced into the flask, followed by increasing the temperature to 330 °C. When the solution became transparent, the temperature was reduced to 150 °C, and a mixture of 0.04 g (0.5 mmol) Se and 4.8 mL TOP solution (mixed in the glovebox, sealed, and dispersed uniformly by ultrasonication) was injected. Subsequently, the system was heated to 250 °C and maintained for 4 h, and then cooled to room temperature naturally. Toluene was added, followed by centrifugation, and the brown precipitate (unreacted CdO) was removed. Methanol was added to the supernatant, followed by centrifugation to produce CdSe QDs as a solid. These dispersion/precipitation steps were repeated twice. For each batch of CdSe QDs ($n_{[Cd2+]} = 0.4$ mmol), 110 μL TGA ($n_{[Cd2+]}$:$n_{TGA} = 1:4$) was dissolved in 15 mL methanol, and its pH value was adjusted to 10 by adding TMAH. The TGA methanol solution was added to the precipitated CdSe QDs and ultra-sonicated for 1 h. The subsequent TGA-capped CdSe QDs were purified by two cycles of precipitation with ethyl acetate/dispersion in methanol. TGA-capped CdSe QDs were stored in the dark.

**Synthesis of CdSe QD gel (electrochemical gelation)**. CdSe QD gels were synthesized by an electrogelation method developed by our group[26]. CdSe QDs were first dispersed into methanol at a concentration of ~36 μM. Nine milliliters of

the CdSe QDs solution was mixed with 1 mL of a 0.1 M $NBu_4PF_6$ methanol solution (electrolyte). A three-electrode setup with an Ag/AgCl electrode in saturated KCl aqueous solution as reference electrode, a Pt foil (~385 mm²) as the counter electrode, and a Pt foil (~240 mm²) as the working electrode was utilized in the electrogelation. An electrode potential of 1.5 V was applied to the working electrode for 1 h using a CHI 650E potentiostat to drive the electrogelation. Before electrogelation, the Pt foils were electrochemically polished in 0.5 M $H_2SO_4$ aqueous solution by running cyclic voltammograms between 1.1 and −0.23 V at a scan rate of 0.1 V/s for 500 cycles, followed by washing with DI water and methanol. After gelation, the CdSe gel was washed with methanol six times and stored in methanol in the dark.

**Synthesis of $Pb_xCd_{1−x}Se$ QD gel (cation exchange)**. $Pb_xCd_{1−x}Se$ QD gels were synthesized by immersing the CdSe QD wet gel in solutions of Pb($NO_3$)$_2$. Different amounts of Pb($NO_3$)$_2$ were dissolved in a mixture of methanol and DI water ($V_{methanol}/V_{DI\ water} = 3:1$) to obtain Pb($NO_3$)$_2$ solutions with various concentrations: 3, 6, 12.5, 25, 50, 150, to 750 mM. Two milliliters of Pb($NO_3$)$_2$ solution was added to 1 mL of CdSe wet gel ($n_{[Cd2+]} = 0.05$ mmol) to produce the $Pb_xCd_{1−x}Se$ wet gel. After 10 min, the supernatant was carefully removed by pipette without disturbing the gel. Then, a mixture of methanol and DI water ($V_{methanol}/V_{DI\ water} = 3:1$) was used to wash the gel. After 10 min, the supernatant was discarded. The purification procedure was repeated ten times and the prepared $Pb_xCd_{1−x}Se$ QD gels were stored in methanol in the dark

**Preparation of $Pb_xCd_{1−x}Se$ QD aerogels for nitrogen physisorption**. $Pb_xCd_{1−x}Se$ QD gels were subjected to $CO_2$ critical point drying (CPD) to yield the corresponding aerogels. First, the methanol supernatant was carefully removed using a pipette without disturbing the gel at the bottom, and the same amount of acetone was added to the vial to replace methanol. The above procedure was repeated five times per day for 1 week before CPD. Second, $Pb_xCd_{1−x}Se$ wet gel immersed in acetone was dried supercritically using a SPI-DRY model $CO_2$ critical point drier equipped with a recirculating water bath (ISOTEMP 10065). The solvent was completely exchanged from acetone to liquid $CO_2$ at 18 °C. The temperature was increased to 37 °C to drive $CO_2$ supercritical. After 30 min, the pressure is slowly released and the resulting $Pb_xCd_{1−x}Se$ aerogel was stored in the dark.

**Gas sensor fabrication and testing**. The sensor substrate is an electrical insulator comprising a sintered alumina plate equipped with interdigitated Pt electrodes (the distance between adjacent electrodes is 0.4 mm)[26]. Before use, the sensor substrates were cleaned by ultrasonication in DI water and methanol. Ten microliters $Pb_xCd_{1−x}Se$ wet gel (alcogel) was drop-casted on the sensor substrates and dried under ambient conditions to produce xerogel films.

The gas-sensing testing of $Pb_xCd_{1−x}Se$ QD gels was performed using a homebuilt apparatus[26] composed of five parts: gas tanks providing air and test gases (Airgas Co., Ltd), mass flow controllers for regulating the flow rate of gases (Bronkhorst), a home-made Teflon chamber where the sensor was housed, a data acquisition card for collecting electrical resistance changes in real time (Keysight/Agilent 34972A LXI), and a computer for data storage. Before testing, all the gel sensors were aged in air by flowing synthetic air (21%$O_2$ + 79%$N_2$, Airgas Co. Ltd) for 8 h until the resistance is stable. During the tests, the target gas was mixed with air to obtain the desired concentration of test gas in the air. The total flow rate was maintained at 2000 sccm, and the relative humidity of the chamber was kept at 50%.

**Wireless portable device fabrication and test**. The sensor substrates used in the wireless portable device were prepared using a standard photolithography process as illustrated in Supplementary Fig. 16. Briefly, Si wafers with a 500-nm-thick oxide layer were cleaned and baked at 120 °C for 5 min followed by spin coating of the LOR 10B photoresist at 4000 r.p.m. for 45 s, then baked again at 190 °C for 5 min. After cooling, the Shipley photoresist was spin-coated at 4000 r.p.m. for 30 s, and then baked at 115 °C for 2 min. Later, the photoresist film was exposed to 350 nm UV light, broad-band 20 mW/cm² for 5.5 s. Development was done by rinsing in AZ-726 developer for 25 s. Metal films of 10 nm Cr followed by 200 nm Au were deposited by PVD and lifted off by soaking in PG remover overnight at room temperature. All photolithography steps were carried out at the Lurie Nanofabrication Facility at the University of Michigan. The geometry and dimensions of the sensor electrodes are shown in Supplementary Fig. 17.

The design of the wireless portable device is shown in Supplementary Fig. 25. The readout signal was calculated from the electrical resistance change of the device using the Adafruit Feather 32u4 Bluefruit microcontroller powered with a 3.3 V battery. A voltage divider was used to measure the resistance (Supplementary Fig. 25). The microcontroller converts the voltage ($V_{in}$) between the analog input pin (A0) and the ground pin (GND) to a digital value between 0 and 1023, corresponding to voltage values between 0 V and $V_c = 3.3$ V (the voltage of the power supply). The value of the device resistance ($R_{device}$) is calculated using Eq. (1).

$$R_{device} = R_{ref} \frac{V_{in}}{V_c - V_{in}} \qquad (1)$$

The gas concentration value ($C_{gas}$) is calculated from the characteristic graph ($R_{sensor}$ vs $C_{gas}$) using the following formula: $C_{gas} = A \times R_{device} + B$, where $A$ and $B$

are determined experimentally. The measured gas concentration value is displayed on an organic light-emitting diode display connected to the microcontroller. The microcontroller is programmed with a code written using the Arduino development environment and uploaded via USB.

**Characterization and measurements**. HAADF-STEM images were taken using a JEOL3100R05 Double Cs Corrected S/TEM operated at 300 kV, with a collection angle of 59–200 mrad. The EDS mapping was carried out on a Thermo Fisher Scientific Talos F200X S/TEM equipped with a Super-X EDS detector. The TEM specimens were prepared by drop-casting wet gel onto carbon-coated 200-mesh Cu grids. The particle size distribution was estimated using Nano Measurer 1.2 software. The crystalline phase was characterized by PXRD using a Bruker D2 Phaser diffractometer. PXRD patterns were analyzed by comparison to the powder diffraction file database of the International Center for Diffraction Data. The chemical state and element ratio were analyzed by X-ray photoelectron spectroscopy (Thermo Fisher Scientific NEXSA UV and X-ray Photoelectron Spectrometer). XPS peaks were fitted using a composite function (30% Lorentzian + 70% Gaussian) and calibrated according to the C1s peak at 284.8 eV via the Avantage software. The elementary ratio of Pb/Cd was also identified by ICP-MS (Agilent 7700x ICP-MS). The surface area and pore size determined from nitrogen physisorption data (Micrometrics ASAP 2020 analyzer) using the BET and Barrett–Joyner–Halenda models. The surface and cross-section morphology of the xerogel film were analyzed by field-emission scanning electron microscopy (JEOL JSM 7600F SEM).

**Theoretical calculations**. All calculations were performed via the Vienna Ab Initio simulation package utilizing DFT[68,69]. Core electrons were described using the projected-augmented wave (PAW) method[70]. The Kohn–Sham wave functions were expanded on a plane-wave basis with a kinetic energy cutoff of 400 eV to describe the valence electrons. The generalized gradient approximation using the Perdew–Burke–Ernzerhof functional was employed to evaluate the exchange-correlation energy[71]. The crystal structure in the phase separation section was prepared using $(3 \times 6 \times 3)$ and $(3 \times 3 \times 3)$ supercells for the hexagonal and cubic phases, respectively. The Monkhorst–Pack scheme was employed to sample the Brillouin zone using $1 \times 1 \times 1$ k-point grid for geometry optimization[72]. Geometry was considered optimized when the force on each atom was <0.02 eV/Å.

The CdSe(100) and PbSe(100) surfaces were modeled with a four-layer $(3 \times 2)$ hexagonal CdSe(100) supercell and a two-layer $(3 \times 2)$ cubic PbSe(100) supercell, respectively. The bottom two layers of CdSe(100) surface and bottom one layer of PbSe(100) surface were kept frozen, while the other layers and adsorbed gas molecules were set free to relax. A vacuum space >15 Å was added to all surface models to ensure no appreciable interaction between periodic images. The Monkhorst–Pack scheme was employed to sample the Brillouin zone using a $3 \times 3 \times 1$ k-point grid for atomic structure optimization and a $7 \times 7 \times 1$ k-point grid for refining electronic structures. Structural optimization was performed until the force on each atom was <0.025 eV/Å. Bader charge analysis was used to decompose the charge density into volumes around atoms[73].

**Machine learning**. As shown in Supplementary Fig. 10, 47 $Pb_xCd_{1-x}Se$ cubic supercells (35 for hexagonal, $0 < x < 1$) were first generated randomly and used as the initial training database after geometry optimization. An automated ML package, TPOT[74], was employed to optimize the ML regression method and hyperparameters for $E_f$ prediction. Based on the prediction of ML surrogate model, the 50000-step Metropolis Monte Carlo (MC) simulation was used to explore the $Pb_xCd_{1-x}Se$ configuration for the most stable structures at each composition (Supplementary Figs. 11 and 12). These newly searched configurations were then labeled with DFT calculations and added to the training database to refine the ML surrogate model. Such an active learning loop of training-MC-DFT was repeated until the prediction accuracy criteria were met. The last batch of the most stable configurations at each composition was then used to construct the thermodynamic convex hull. Our scheme can be understood as an exploitation dedicated version of the Bayesian optimization (BO). Exhilaratingly, the ML surrogate model exhibited high confidence (cubic: MAE = 0.0028, $R^2 = 0.96$; hexagonal: 0.0014, $R^2 = 0.97$) for predicting the most stable candidate suggested by MC after iterations (Supplementary Figs. 13 and 14). The features were extracted by counting the numbers of different coordination pairs in the metal ion matrix, i.e., the number of Pb or Cd atoms with different numbers of Pb in the first nearest neighbor sphere.

## Data availability
The data that support the findings of this study are available from the authors on reasonable request. Source data are provided with this paper.

## Code availability
The active learning code (CAMkit) used to construct the thermodynamic convex hull for $Pb_xCd_{1-x}Se$ is still under further development, but a beta version is available from the corresponding author Z.L. upon reasonable request.

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

## Acknowledgements

This work was financially supported by the start-up funds of L.L. from Wayne State University, the start-up funds of L.Z. from Tsinghua University, a National Science Foundation (NSF) grant of S.L.B. (grant number CHE-1709776), Air Force Office of Scientific Research (AFOSR) of G.M. (grant number FA2386-20-1-4077), and the Camille Dreyfus Teacher-Scholar Award of E.N. This work utilized an XPS that is partially funded by NSF grant #1849578, a JEOL-2010 TEM supported by NSF grant #0216084, the PXRD Facility supported by NSF grant #1427926, and the 7-BM of the National Synchrotron Light Source II, a U.S. Department of Energy (DOE) Office of Science User Facility operated for the DOE Office of Science by Brookhaven National Laboratory under Contract No. DE-SC0012704 and a JEOL3100R05 at the Michigan Center for Materials Characterization, the University of Michigan, funded by NSF grant # 723032. We thank Dr. Sen Zhang and his student at the University of Virginia for their help with the XAS measurements.

## Author contributions

X.G. and L.L. conceived this project, performed most of the experiments, and wrote the paper. S.L. and L.Z. did theoretical calculations and co-wrote the paper. L.M.-V. and S.L.B. prepared the QDs, participated in part of the characterization, and edited the paper. T.M. carried out the HAADF-STEM and EDS characterization, as well as the analysis of single lead ions according to relevant images. M.K. and G.M. produced the wireless portable device and wrote the device fabrication part. B.W. and E.N. performed the BET test and analysis. L.M. performed the XAS measurements. C.C.H.-R. and A.S. synthesized some samples.

## Competing interests

The authors declare no competing interests.
