## [Peer Review File · Nature Communications]

REVIEWER COMMENTS

Reviewer #1 (Remarks to the Author):

This article describes the development of an NO₂ sensor based on cation-exchanged PbCdSe quantum dot gels. The authors found that exchanges with low lead percentages results atomically dispersed Pb surface sites on the CdSe nanocrystals and that these Pb atoms greatly enhance the detector's performance. Overall, this paper is well written and the authors have performed a thorough balance of experiment and theory to support their claims. Further the authors do a good job not over selling the identification of Pb atoms in their HAADF-STEM images, as the Z-contrast can be difficult to interpret for non-flat samples. The only concern that I have is that a control experiment is needed which shows that a similar loading of the lead cations in the matrix does not have a the same affect, thus showing that the LOD improvement is a direct result of the Pb-atoms cation exchanged. Lastly, the authors show STEM-EDS maps but do not show the integrated spectrum. This should be included, at least, in the supporting information to give the reader a sense of the signal-to-noise associated with each element shown. After these additions, this manuscript should be acceptable for publication in Nature Communications.

Reviewer #2 (Remarks to the Author):

The authors synthesized the QD gel using the electro-oxidative removal of the protecting thiolate ligands technique based on the CdSe quantum dots synthesized by the modified hot-injection method, and finally modulated the composition of Pb and Cd via a cation exchange process. The trade-off correlation between the sensor sensitivity, response time, and recovery time of the NO₂ sensor fabricated based on the thus produced PbCdSe QD gel was optimized. From the novelty and importance of this paper's motivation, sensor material synthesis, and sensor characteristics results, I believe that this paper is good enough to be published in Nature Communication, so I recommend that it be accepted after a minor revision.

1. Please organize the data in Figure 4c into a table and display it in numerical form. From this picture, it is not sure whether "the sensor response to 1.32 ppm NO₂ has already increased by 1300% as x is slightly changed from 0 to 0.09, whereas it only increases by another 486% when x is further increased to 1.0"

2. As to the sensor fabrication process, it was fabricated by drop-casting PbCdSe QD gel on a metal IDE pattern prepared in advance by micro-patterning and drying in air. It is expected that the electrical resistance between the PbCdSe QD gel and the metal pattern might be quite large unless an additional heat treatment process is performed. It is recommended to measure the specific contact resistance between the IDE and PbCdSe QD gels.

Reviewer #3 (Remarks to the Author):

The study performed by the Brock-Zhang-Luo Groups focuses on the preparation of a highly effective NO₂ sensor based on semiconductor nanocrystal gel structures. The Authors fabricate the sensor via deposition of partially exchanged CdSe QD gel via Pb²⁺ ions using a three-step procedure. As the key novelty, the Authors found that an ultra-sensitive, fast and exceptionally stable NO₂ sensor can be built up using PbxCd1-xSe QD gels.

Additionally, the structure-property-mechanism triangle is also greatly investigated with state-of-the-art measurement as well as simulation techniques (including machine learning, which is impressive).

The manuscript is well-written, well-structured, and its overall quality is outstanding. Although the structure of the gels and the nature/effect of the Pb in the CdSe matrix is in the spotlight of the paper and studied from different aspects, some major concerns need to be addressed and the following points should be clarified before publication:

- As a general question: what is the benefit of using gel structures, if the final step of the preparation includes a simple drop casting and ambient drying of the initially highly porous gel?
- Although the Authors claim that the deposited xerogel (on the top of the electrode) 'exhibits a highly porous surface morphology', it is not fully supported by the Supplementary Figure 14 and 15, where a significantly collapsed structure is shown. This structure possesses solely macropores, or the retained mesoporosity (shown in the TEM images of the aerogels) is not visible. Please comment on this.
- Closely connected to the previous points, the obtained highly collapsed xerogel structure stands closer to a QD multilayer than to the original aerogel structures. Please give more explanations, how this collapse influences the structure-sensing correlation (considering the fact that the BET surface areas dramatically decrease upon drying which also affects the availability of the active surface sites during the sensing). This must be addressed and explained in the main text for clarity.
- Moreover, these drying effects (capillary forces) could be greatly suppressed via using more volatile solvents (acetogels, alcogels, etc). Why did the Authors use hydrogels (if the term 'wet gel' refers to hydrogel)?
- What is the benefit to perform the gelation on the already formed gel structures instead of on the QDs prior to the gelation? This is a crucial key step in the fabrication procedure; thus, more explanation is needed. Would not the pre-gelation cation exchange be more controllable in terms of composition and kinetics? How does the highly porous gel structure influence these aspects of the cation exchange. Effect on homogeneity should also be mentioned.
- The Authors employed hexagonal CdSe as source QDs but reveal a dramatic effect of the (partial) cation exchange on the crystallinity, size of the QDs which they attribute to the structural disruption due to the rapid exchange. Two questions arise here: (i) why do not you use cubic CdSe QDs, and (ii) how could the driving force be reduced significantly to ensure the preservation of the QD morphology even upon full exchange?
- How can the above-mentioned structural disruption be explained based on the findings in REF48: 'The ion-exchange synthesis route can also be successfully employed for other gel systems, e.g., PbSe and CuSe'? In the quoted reference, the possibility of full exchange was shown. Please give an explanation on this.
- Authors shows the shifting and broadening of the XPS peaks in Figure 3B. While the shift goes along with a certain broadening, I agree to the observation that the two forms of Pb (isolated ionic vs cubic phase) are both present simultaneously. How can the ratio of these forms be controlled? This ratio has a dramatic effect on the NO₂ sensing performance as demonstrated in the second part of the paper, thus, it should be determined.
- A more in-detail conclusion from the XANES measurements is needed in the main text, because it is too brief and rather vague.
- I found the term 'isolated Pb site' confusing, thus, I would recommend revisiting the terms and use e.g. ionic sites and segregated sites (or something more distinguishable) to fully separate the atomically dispersed Pb from the Pb in PbSe matrix. This would improve the

readability (especially in the DFT and mechanism parts) and understand the main message: ionic sites are superior compared to segregated PbSe sites from sensing performance point of view.

The novelty is remarkable, the conclusion is supported by the data, and after revision, the paper will definitely have the community's attention. The method is well-described in the experimental section for reproducibility. After addressing the major concerns, the sensing performance warrants the paper to be published in Nature Communications.

Reviewer #4 (Remarks to the Author):

I would recommend authors to clarify the following issues:

1. Authors claim that "Short-term exposure to a high concentration of NO₂ irritates the human respiratory system, causing respiratory distress symptoms such as coughing, wheezing, and difficulty breathing." Sensing strategy authors proposed implies, however, exploitation of toxic heavy metals (Cd and Pb) that can adversely affect the human health in their own way. In many countries, the use of these hazardous materials in electrical and electronic devices is undesirable. Could authors justify somehow utilization of Cd- and Pb-containing materials for environmental sensing?

2. Alumina was used as a sensor substrate. It would be good to add several sentences explaining why did authors choose this substrate? What is a strength of gel adhesion to the surface of the alumina substrate? Is there any interaction between the substrate and PbxCd_{1-x}Se QD gel? Could this interaction potentially affect the sensing performance?

3. On Page 8, authors introduce palladium sites. Probably this is a typo. Otherwise, authors need to explain a role of Pd in more details.

line 23 "For all bimetallic surfaces, their Cd sites adjacent to Pd show stronger adsorption energies and larger"

line 37 "The above findings show that (1) the Cd sites adjacent to Pd cations in PbxCd_{1-x}Se surface lead to"

4. Page 6, line 26-30. Authors wrote "PbxCd_{1-x}Se gel sensors were prepared by drop-casting 10 μL of wet gel onto a sensor substrate patterned with an interdigitated electrode, followed by drying in air. The dried gel film exhibits a highly porous surface morphology (Supplementary Fig. 14) and a thickness of ~3.1 μm (Supplementary Fig. 15)."

Is there any special reason why did authors prepare the gel film of ~3.1 μm thickness? Is this thickness optimal? Will nm-thick gel films do have other sensing performance?

5. As the isolated atomic sites in hexagonal matrix authors considered Pb²⁺ divalent ions. From the theoretical point of view, the presence of charged ions will cause the formation of surface dipole. Did authors apply the dipole correction to compute a total energy of certain system? Should Pb on the surface of an CdSe be treated as a charged ion or as a neutral atom? In other words, the system containing isolated ions was neutral or charged?

Response to Reviewers' Comments

Ms. ID: NCOMMS-21-15323

Title: Isolated metal ion geometries as a strategy for room-temperature NO₂ sensing

Reviewers' comments in normal type

Author response in italics

Verbatim quotes from the revised text are in boldface

Reviewer 1

Comments:

This article describes the development of an NO₂ sensor based on cation-exchanged PbCdSe quantum dot gels. The authors found that exchanges with low lead percentages results atomically dispersed Pb surface sites on the CdSe nanocrystals and that these Pb atoms greatly enhance the detector's performance. Overall, this paper is well written and the authors have performed a thorough balance of experiment and theory to support their claims. Further the authors do a good job not over selling the identification of Pb atoms in their HAADF-STEM images, as the Z-contrast can be difficult to interpret for non-flat samples.

1. The only concern that I have is that a control experiment is needed which shows that a similar loading of the lead cations in the matrix does not have a same affect, thus showing that the LOD improvement is a direct result of the Pb-atoms cation exchanged.

*Author reply: The reviewer raised an excellent point. To address the reviewer's concern, we carried out the following control experiment. 0.005 mmol lead nitrate was mixed with CdSe xerogel (prepared from 1 mL of CdSe wet gel, $n_{[Cd^{2+}]}$ =0.05 mmol) in hexane. Because lead nitrate does not dissolve in nonpolar solvents like hexane, cation exchange will not occur, forming a CdSe gel physically loaded with lead nitrate ($Pb(NO_3)_2@CdSe$ gel) with a Pb: Cd ratio of 0.09:0.91. The XRD patterns of $Pb(NO_3)_2@CdSe$ gel confirmed the presence of $Pb(NO_3)_2$ and CdSe gel (**Figure R1a**). Furthermore, the XPS result in **Figure R1b** shows the Pb 4f 5/2 peak of $Pb(NO_3)_2@CdSe$ gel is located at 144.3 eV, same as the Pb peak of $Pb(NO_3)_2$ rather than atomically dispersed Pb ionic sites (143.5 eV) or Pb^{2+} ions for cubic phase (143.8 eV), further confirming no cation exchange took place under the control experiment conditions, and the CdSe gel solely functioned as a matrix for Pb. We tested the sensing performance of $Pb(NO_3)_2@CdSe$ gel and found its performance was comparable to CdSe gel's, but much worse than $Pb_{0.09}Cd_{0.91}Se$ gel, in terms of LOD, sensor response, response and recovery time (**Figure R1c and Table R1**), indicating the high performance requires the effective cation exchange between Pb ions and Cd in CdSe gel.*

*We added **Figure R1** as **Supplementary Figure 23** to the supplementary information and the following discussion to the main text.*

Page 7: "Note that the loading of $Pb(NO_3)_2$ in a CdSe QD gel matrix without cation exchange (achieved by mixing $Pb(NO_3)_2$ with CdSe QD gel in hexane) does not lead to any sensing performance improvement (Supplementary Fig. 23), suggesting the high performance of $Pb_{0.09}Cd_{0.91}Se$ QD gel is a direct result of cation exchange."

Figure R1. a, XRD patterns of $Pb(NO_3)_2@CdSe$ gel, $CdSe$ QD gel, and $Pb(NO_3)_2$. b, XPS results of the Pb 4f 5/2 region for $Pb(NO_3)_2@CdSe$ gel and $Pb(NO_3)_2$. c, Response–recovery curve of a $Pb(NO_3)_2@CdSe$ gel sensor in response to NO_2 (3 ppb–1.32 ppm) at room temperature. $Pb(NO_3)_2@CdSe$ gel is a $CdSe$ gel physically loaded with $Pb(NO_3)_2$ with a $Pb: Cd$ ratio of 0.09:0.91, prepared by mixing 0.005 mmol $Pb(NO_3)_2$ with $CdSe$ xerogel ($n_{[Cd^{2+}]}=0.05$ mmol) in a nonpolar solvent, hexane, to prevent cation exchange between Pb salt and $CdSe$ QD gel.

Table R1. Comparison of sensing performance of $Pb(NO_3)_2@CdSe$ gel ($Pb: Cd = 0.09:0.91$), $CdSe$ gel and $Pb_{0.09}Cd_{0.91}Se$ gel in response to 1.32 ppm NO_2 at room temperature.

	Sensor response (S , %)	Response time (t_{res} , s)	Recovery time (t_{rec} , s)	Limit of detection (LOD, ppb)
$Pb(NO_3)_2@CdSe$ gel ($Pb: Cd = 0.09:0.91$)	5.7	25	30	440
$CdSe$ QD gel	5.2	24	31	440
$Pb_{0.09}Cd_{0.91}Se$ QD gel	72.8	28	66	3

2. Lastly, the authors show STEM-EDS maps but do not show the integrated spectrum. This should be included, at least, in the supporting information to give the reader a sense of the signal-to-noise associated with each element shown. After these additions, this manuscript should be acceptable for publication in Nature Communications.

Author reply: Great point. The integrated EDS mapping results are shown in **Figure R2**.

We added **Figure R2** as **Supplementary Figure 6** to the supplementary information.

Figure R2. Integrated EDS elemental mappings of a, $Pb_{0.04}Cd_{0.96}Se$; b, $Pb_{0.09}Cd_{0.91}Se$; c, $Pb_{0.17}Cd_{0.83}Se$; d, $Pb_{0.40}Cd_{0.60}Se$ QD gels.

Reviewer 2

Comments:

The authors synthesized the QD gel using the electro-oxidative removal of the protecting thiolate ligands technique based on the CdSe quantum dots synthesized by the modified hot-injection method, and finally modulated the composition of Pb and Cd via a cation exchange process. The trade-off correlation between the sensor sensitivity, response time, and recovery time of the NO₂ sensor fabricated based on the thus produced PbCdSe QD gel was optimized. From the novelty and importance of this paper's motivation, sensor material synthesis, and sensor characteristics results, I believe that this paper is good enough to be published in Nature Communication, so I recommend that it be accepted after a minor revision.

1. Please organize the data in Figure 4c into a table and display it in numerical form. From this picture, it is not sure whether "the sensor response to 1.32 ppm NO₂ has already increased by 1300% as x is slightly changed from 0 to 0.09, whereas it only increases by another 486% when x is further increased to 1.0".

*Author reply: The reviewer brought up an excellent point. We summarized the data from Figure 4c in numerical form and tabulated them in **Table R2**.*

*We added **Table R2** as **Supplementary Table 6** to the supplementary information.*

Table R2. Sensor response (S), response time (t_{res}), and recovery time (t_{rec}) of $Pb_xCd_{1-x}Se$ QD gels towards 1.32 ppm NO_2 as a function of x .

x in $Pb_xCd_{1-x}Se$	Sensor response (S , %)	Response time (t_{res} , s)	Recovery time (t_{rec} , s)
0	5.2	24	31
0.003	6.8	25	34
0.02	44.7	26	46
0.04	52.4	27	53
0.09	72.8	28	66
0.17	83.5	29	85
0.4	93.7	30	150
1	98.1	30	240

2. As to the sensor fabrication process, it was fabricated by drop-casting PbCdSe QD gel on a metal IDE pattern prepared in advance by micro-patterning and drying in air. It is expected that the electrical resistance between the PbCdSe QD gel and the metal pattern might be quite large unless an additional heat treatment process is performed. It is recommended to measure the specific contact resistance between the IDE and PbCdSe QD gels.

Author reply: An excellent point! We measured the contact resistance between $Pb_{0.09}Cd_{0.91}Se$ gel and the metal electrodes following a procedure in the literature.¹ Briefly, the measured resistance ($R_{measured}$) is the sum of the contact resistance ($2R_{contact}$) at the two electrodes and the resistance of $Pb_{0.09}Cd_{0.91}Se$ gel (R_{gel}), as expressed by:

$$R_{measured} = 2R_{contact} + R_{gel} = 2R_{contact} + \rho L/A$$

*where ρ and A are the resistivity and cross-sectional area of $Pb_{0.09}Cd_{0.91}Se$ gel, respectively, and L is the distance between two measuring electrodes. The value of $R_{contact}$ can be obtained from the intercept of the linear plot of $R_{measured}$ against L . Experimentally, we deposited a uniform $Pb_{0.09}Cd_{0.91}Se$ gel layer onto a substrate patterned with four electrodes (P1 to P4 in **Figure R3a**). By selecting different combinations of the four electrodes, we varied the L -value (for example, L for the P1-P2 electrode pair is 200 μm) and measured the corresponding $R_{measured}$. **Figure R3b** shows the plot of $R_{measured}$ vs L , giving an intercept of (0.031 $M \Omega$) or a contact resistance of $\sim 0.015 M \Omega$, which is two orders of magnitude smaller than the few $M \Omega$ resistance of $Pb_{0.09}Cd_{0.91}Se$ gel and, thus, is negligible.*

*We added **Figure R3** as **Supplementary Figure 21** to the supplementary information and also added the following discussion to the main text.*

Page 7: “The contact resistance between QD gel film and electrodes is ~ 2 orders of magnitude smaller than the gel film’s resistance and, thus, is negligible (Supplementary Fig. 21).”

Figure R3. a, Schematic diagram of the sensor substrate design for measuring contact resistance for the $Pb_{0.09}Cd_{0.91}Se$ gel. b, Extrapolated line fitting for determining the contact resistance from the plot of total resistance ($R_{measured}$) versus the distance between two electrodes (L) for each unique combination (e.g., P1-P2, P1-P3, etc.) according to $R_{measured} = 2R_{contact} + R_{gel} = 2R_{contact} + \rho L/A$, where ρ is the gel resistivity and A is the cross-sectional area. The y-axis intercept (0.031 $M\Omega$) is twice the contact resistance.

Reviewer 3

Comments:

The study performed by the Brock-Zhang-Luo Groups focuses on the preparation of a highly effective NO_2 sensor based on semiconductor nanocrystal gel structures. The Authors fabricate the sensor via deposition of partially exchanged CdSe QD gel via Pb^{2+} ions using a three-step procedure. As the key novelty, the Authors found that an ultra-sensitive, fast and exceptionally stable NO_2 sensor can be built up using $Pb_xCd_{1-x}Se$ QD gels. Additionally, the structure-property-mechanism triangle is also greatly investigated with state-of-the-art measurement as well as simulation techniques (including machine learning, which is impressive). The manuscript is well-written, well-structured, and its overall quality is outstanding. Although the structure of the gels and the nature/effect of the Pb in the CdSe matrix is in the spotlight of the paper and studied from different aspects, some major concerns need to be addressed and the following points should be clarified before publication:

1. As a general question: what is the benefit of using gel structures, if the final step of the preparation includes a simple drop casting and ambient drying of the initially highly porous gel?

Author reply: A good question. During the ambient drying of the highly porous wet gel, we did lose some porosity. However, apart from high porosity, the QD gel after ambient drying still preserves several other advantages of gels for gas sensing, including small crystallite size (high surface-to-volume ratio), connected network (facilitated electronic communication), and rich chemistry (easy surface modification), as we explained in the introduction. Besides, compared to QDs, the organic ligands are partially or entirely removed from particle surface in QD gel, which substantially increases the number of active surface sites for gas molecules to bind, improving the sensing performance.

We added the following discussion to the introduction (paragraph 2).

Page 2: “Moreover, the process of gelation has the added benefit of partially or entirely stripping organic ligands from the particle surface, thereby substantially increasing the number of active surface sites for gas molecules to bind, improving the sensing performance.”

2. Although the Authors claim that the deposited xerogel (on the top of the electrode) ‘exhibits a highly porous surface morphology’, it is not fully supported by the Supplementary Figure 14 and 15, where a significantly collapsed structure is shown. This structure possesses solely macropores, or the retained mesoporosity (shown in the TEM images of the aerogels) is not visible. Please comment on this.

*Author reply: The reviewer brought up a great point. Limited by the resolution of FE-STEM due to the limited conductivity of $Pb_{0.09}Cd_{0.91}Se$ gel (even after Au sputtering), the mesopores of the gel are not visible. To address the reviewer’s concern, we sampled the gel film by touching its surface using a TEM grid to transfer the solid gel sample to the grid and then took the TEM image, which shows the mesoporous structure was still retained despite partial pore collapse during the ambient drying (**Figure R4**).*

*We added **Figure R4** as **Supplementary Figure 18c** to the supplementary information.*

Figure R4. TEM image of $Pb_{0.09}Cd_{0.91}Se$ gel sensor film.

3. Closely connected to the previous points, the obtained highly collapsed xerogel structure stands closer to a QD multilayer than to the original aerogel structures. Please give more explanations, how this collapse influences the structure-sensing correlation (considering the fact that the BET surface areas dramatically decrease upon drying which also affects the availability of the active surface sites during the sensing). This must be addressed and explained in the main text for clarity.

*Author reply: The reviewer brought up a great point. One would expect the highly collapsed xerogel structure stands closer to a QD multilayer. However, in our case, even though the surface area of $Pb_{0.09}Cd_{0.91}Se$ gel decreased significantly during the ambient drying relatively to the aerogel according to the BET measurements (**Figure R5a**, **Table R3**), the mesoporous structure is still retained (**Figure R4** and **Figure R5b**, **Table R3**) possibly because of the volatile solvent (methanol) in the wet gel.*

Ideally, aerogel should be the better choice than ambient-dried wet gel for gas sensing, but unfortunately, it imposes a practical challenge in preparing a robust and uniform aerogel thin film on a sensor substrate because aerogel does not bind well with the sensor substrate and would be easily peeled off by a gentle gas flow during sensing tests. To address this challenge, we attempted first mixing the aerogel with methanol to make a slurry, then depositing it onto the sensor substrate and drying the sensor in air to prepare $Pb_{0.09}Cd_{0.91}Se$ A-xerogel sensor with a thickness of $3.4\ \mu m$ (Figure R5c), wishing to reduce the pore collapse. However, it turned out the A-xerogel film exhibited a similar mesoporous structure with partial pore collapse as the xerogel film currently used (Figure R5 and Table R3). In addition, the sensing performance of A-xerogel is also comparable to that of xerogel (Figure R5f and Table R4).

We added Figure R5 as Supplementary Figure 20, Table R3 as Supplementary Table 2 in the supplementary information and also added the following discussion to the main text.

Page 7: “The mesoporous structure of gel was still retained in the xerogel film, although the surface area decreased significantly compared to the aerogel due to partial pore collapse during ambient drying (Supplementary Fig. 20 and Supplementary Table 2).”

Figure R5. a, Nitrogen adsorption-desorption isotherms; b, Barrett–Joyner–Halenda pore size distributions of $Pb_{0.09}Cd_{0.91}Se$ xerogel and A-xerogel. c, Cross-section FE-SEM image of $Pb_{0.09}Cd_{0.91}Se$ A-xerogel sensor; d-e, FE-SEM and TEM image of the $Pb_{0.09}Cd_{0.91}Se$ A-xerogel sensor film; f, Response-recovery curve of $Pb_{0.09}Cd_{0.91}Se$ xerogel and A-xerogel sensors in response to 1.32 ppm NO_2 at room temperature.

Table R3. BET data of $Pb_{0.09}Cd_{0.91}Se$ xerogel and A-xerogel. Xerogel was prepared by drying $Pb_{0.09}Cd_{0.91}Se$ wet gel under ambient conditions, while A-xerogel was prepared by mixing aerogel with methanol to make a slurry then drying under ambient conditions.

Sample	BET surface area (m^2/g)	BJH average pore diameter (nm)	BJH cumulative pore volume (cm^3/g)
$Pb_{0.09}Cd_{0.91}Se$	30.4	3.0 (adsorption isotherm)	0.2 (adsorption isotherm)

xerogel		2.9 (desorption isotherm)	0.3 (desorption isotherm)
Pb_{0.09}Cd_{0.91}Se	35.1	6.4 (adsorption isotherm)	0.1 (adsorption isotherm)
A-xerogel		5.4 (desorption isotherm)	0.1 (desorption isotherm)

Table R4. Comparison of sensing performance for *Pb_{0.09}Cd_{0.91}Se* xerogel and *A-xerogel* sensors towards 1.32 ppm *NO₂* at room temperature.

	Sensor response (S , %)	Response time (t_{res} , s)	Recovery time (t_{rec} , s)
Pb_{0.09}Cd_{0.91}Se xerogel	72.8	28	66
Pb_{0.09}Cd_{0.91}Se A-xerogel	74.9	27	63

4. Moreover, these drying effects (capillary forces) could be greatly suppressed via using more volatile solvents (acetogels, alcogels, etc). Why did the Authors use hydrogels (if the term ‘wet gel’ refers to hydrogel)?

Author reply: An excellent point. The wet gels in our work are, in fact, alcogels and do not refer to hydrogel because the wet gels were stored in methanol. To further clarify it, we added a short note next to “wet gel” on page 7 in the main text as below.

“...10 μ L of wet gel (or alcogel as the solvent is methanol)...”

5. What is the benefit to perform the gelation on the already formed gel structures instead of on the QDs prior to the gelation? This is a crucial key step in the fabrication procedure; thus, more explanation is needed. Would not the pre-gelation cation exchange be more controllable in terms of composition and kinetics? How does the highly porous gel structure influence these aspects of the cation exchange. Effect on homogeneity should also be mentioned.

Author reply: We thank the reviewer for the excellent questions.

We designed the current material synthesis procedure with the following considerations in mind.

- (1) *Pre-gelation cation exchange at the nanoparticle (NP) stage is feasible and possibly more controllable but requires more complex procedures than cation exchange at the gel stage. Due to the similarity of Cd^{2+} and Pb^{2+} in their valency, hardness, and electronegativity, it is challenging to directly convert $CdS(Se)$ NPs to $PbS(Se)$ NPs in the presence of capping agents on the NP surface.²⁻⁴ As a result, such conversion often requires an intermediate conversion of $CdS(Se)$ to $Cu_2S(Se)$ or $Ag_2S(Se)$ to produce the final $PbS(Se)$.²⁻⁴ In contrast, because the capping agents used to stabilize QDs are nearly entirely removed during gelation,⁵⁻⁶ the surface of QD gels is much more accessible to the incoming Pb^{2+} cations. Once the surface Cd^{2+} cations are exchanged, the concentration gradient of Cd created generates a reaction potential (Donnan potential)⁷ in the gel network, enabling the cation exchange to take place under mild conditions.*
- (2) *Gelation kinetics are known to be slow for cubic polymorphs, and the thermodynamic*

structure of PbSe is cubic, whereas CdSe can be prepared as hexagonal. Thus, as more Pb is incorporated into the hexagonal (wurtzite) CdSe, phase segregation will occur, leading to cubic and hexagonal domains, each exhibiting different kinetics of oxidative assembly. Indeed, the quality and homogeneity of PbSe gels produced from the gelation of PbSe quantum dots are quite low. Thus, our typical approach to PbSe gels is by CdSe gel formation followed by ion exchange.

- (3) The porous gel structure is favorable because its surface area is more accessible to the incoming cations relative to a dense film, enabling them to be distributed uniformly across the gel structure⁵⁻⁶ as evidenced by the HAADF-STEM and EDS mapping results of $Pb_xCd_{1-x}Se$ gels at $x < 0.17$.

We added the following

Page 3: “Based on prior work demonstrating (1) slow gelation kinetics for cubic polymorphs (i.e., PbSe) relative to hexagonal polymorphs,^{52, 53} and (2) facilitated cation-exchange on ligand-stripped surfaces, our strategy for targeting $Pb_xCd_{1-x}Se$ QD gels involves initial synthesis of hexagonal (wurtzite) CdSe, subsequent gelation (induced by ligand stripping), and ultimately ion-exchange of Cd^{2+} for Pb^{2+} ”

6. The Authors employed hexagonal CdSe as source QDs but reveal a dramatic effect of the (partial) cation exchange on the crystallinity, size of the QDs which they attribute to the structural disruption due to the rapid exchange. Two questions arise here: (i) why do not you use cubic CdSe QDs, and (ii) how could the driving force be reduced significantly to ensure the preservation of the QD morphology even upon full exchange?

Author reply: We thank the reviewers for the two excellent questions.

With respect to point (i), we did not use cubic CdSe for the reason described in our response to point 5, above: the gelation kinetics are slow for the cubic phase and the resultant gel monolith is weak and compact relative to the hexagonal. This is because of the differences in the surface energies of cubic and hexagonal phases.⁸⁻⁹ For example, the aggregation of 4-fluorothiophenolate-capped hexagonal CdS QDs took place within several minutes after introducing an oxidant, whereas oxidative assembly of cubic CdS QDs capped with the same ligand occurred over several hours and formed less rigid, but more dense monoliths. The synthesis method that takes less time and produces better-formed monoliths is preferred.

The changes made in response to point 5 above also address this issue, and are replicated below.

Page 3: “Based on prior work demonstrating (1) slow gelation kinetics for cubic polymorphs (i.e., PbSe) relative to hexagonal polymorphs,^{52, 53} and (2) facilitated cation-exchange on ligand-stripped surfaces, our strategy for targeting $Pb_xCd_{1-x}Se$ QD gels involves initial synthesis of hexagonal (wurtzite) CdSe, subsequent gelation (induced by ligand stripping), and ultimately ion-exchange of Cd^{2+} for Pb^{2+} ”

As to point (ii), we agree that appropriate modification of driving force should result in less ripening during the ion exchange, and this should be possible by modifying the solvent system to alter the solvation energies of Cd^{2+} and Pb^{2+} and using lower Pb^{2+} concentrations. However, we

do consider this effort to be beyond the scope of the present paper.

To address this point, the following text has been added:

Page 4: “This ripening may be due to structural disruption from the rapid cation exchange between Cd^{2+} and Pb^{2+} under forcing conditions created by high Pb^{2+} concentrations. This can potentially be remedied by slowing the kinetics, e.g., by adjusting the solvent (playing off differences in solvation energy for Cd^{2+} and Pb^{2+}) and/or using a lower ionic concentration of the exchanging ion. For the present study, we selected to focus on $Pb_xCd_{1-x}Se$ QD gels with $x \leq 0.40$, to minimize potential contributions from morphological changes on the observed sensing performance.”

7. How can the above-mentioned structural disruption be explained based on the findings in REF48: ‘The ion-exchange synthesis route can also be successfully employed for other gel systems, e.g., PbSe and CuSe’? In the quoted reference, the possibility of full exchange was shown. Please give an explanation on this.

Author reply: We thank the reviewer for asking this question. In Ref. 48, the cation exchange condition they used is different from this work. In this work, a much lower concentration of $Pb(NO_3)_2$ in methanol solution was used to slow down the cation exchange, which reduced the gel structural disruption.

8. Authors shows the shifting and broadening of the XPS peaks in Figure 3B. While the shift goes along with a certain broadening, I agree to the observation that the two forms of Pb (isolated ionic vs cubic phase) are both present simultaneously. How can the ratio of these forms be controlled? This ratio has a dramatic effect on the NO_2 sensing performance as demonstrated in the second part of the paper, thus, it should be determined.

*Author reply: We thank the reviewer for the great suggestion. We fit the Pb 4f 5/2 spectra with two peaks using a composite function of 30% Lorentzian+70% Gaussian (**Figure R6** and **Table R5**). The peak in green at lower binding energies (~143.5 eV) belongs to the isolated Pb ionic sites, and the other one in pink at higher binding energies (~143.8 eV) to the Pb in cubic phase PbSe. The XPS fitting data show that $Pb_{0.17}Cd_{0.83}Se$ and $Pb_{0.4}Cd_{0.6}Se$ gels have two forms of Pb: atomically dispersed Pb ionic sites and Pb in cubic phase PbSe, whereas other gels only have one of the two, consistent with the results of HAADF-STEM, EDS mapping, and XRD characterizations.*

*Due to the significant difference in NO_2 adsorption energy between the cubic phase PbSe and atomically dispersed Pb ionic sites (**Figure 5a**), once the cubic phase appears, regardless of the ratio between the two forms of Pb, the sensing performance will rapidly deteriorate (**Table R6**). As a result, one should avoid cubic phase PbSe rather than control the ratio between two forms of Pb.*

*We added **Figure R7** as **Supplementary Figure 7**, and **Table R5** as **Supplementary Table 3** to the supplementary information.*

Figure R6. XPS peak analysis of Pb 4f 5/2 region for $Pb_xCd_{1-x}Se$ QD gels with $x=0.02, 0.04, 0.09, 0.17, 0.40$ and 1.0 . Fitting was performed using a composite function (30% Lorentzian + 70% Gaussian).

Table R5. XPS peak analysis of Pb 4f 5/2 region for $Pb_xCd_{1-x}Se$ QD gels with $x=0.02, 0.04, 0.09, 0.17, 0.40$ and 1.0 . Fitting was performed using a composite function (30% Lorentzian + 70% Gaussian).

		Atomically dispersed sites	Cubic phase	Atomically dispersed sites (%)	Cubic phase (%)
$Pb_{0.02}Cd_{0.98}Se$	Binding energy (eV)	143.47	143.82	100	0
	Area ratio	1	0		
$Pb_{0.04}Cd_{0.96}Se$	Binding energy (eV)	143.52	143.82	100	0
	Area ratio	1	0		
$Pb_{0.09}Cd_{0.91}Se$	Binding energy (eV)	143.50	143.81	100	0
	Area ratio	1	0		
$Pb_{0.17}Cd_{0.83}Se$	Binding energy (eV)	143.46	143.84	44.4	55.6
	Area ratio	0.80	1		
$Pb_{0.40}Cd_{0.60}Se$	Binding energy (eV)	143.45	143.78	16.0	84.0
	Area ratio	0.19	1		
$PbSe$	Binding energy (eV)	143.49	143.77	0	100
	Area ratio	0	1		

Table R6. Sensor response (S), and recovery time (t_{rec}) of $Pb_xCd_{1-x}Se$ QD gels towards 1.32 ppm NO_2 as a function of x .

x in $Pb_xCd_{1-x}Se$	Sensor response (S , %)	Increase of S relative to $Pb_{0.09}Cd_{0.83}Se$ (%)	Recovery time (t_{rec} , s)	Increase of t_{rec} relative to $Pb_{0.09}Cd_{0.83}Se$ (%)
0.09 (atomically dispersed)	72.8	0	66	0
0.17 (atomically dispersed, cubic phase)	83.5	14.7	85	28.8
0.4 (atomically dispersed, cubic phase)	93.7	28.7	150	127.3

9. A more in-detail conclusion from the XANES measurements is needed in the main text, because it is too brief and rather vague.

*Author reply: We thank the reviewer for the excellent suggestion. Because of the significant X-ray absorption interference between the Se K-edge (12657.8 eV) and the Pb L-edge (13035.2 eV), it is impossible to draw a definitive conclusion from the XAS results of Pb and Se. However, we analyzed the XAS data for the Cd K-edge and found that the chemical environment for Cd in $Pb_xCd_{1-x}Se$ QD gels was similar. Therefore, we expanded the discussion of the XAS results as follows and added the EXAFS data fitting results as **Supplementary Table 4**.*

Page 6: "...As shown in Supplementary Fig. 8, X-ray absorption near edge structure (XANES) of the Cd K-edge for the $Pb_xCd_{1-x}Se$ and CdSe QD gels did not show any obvious differences in absorption energy: the first derivative peak positions are CdSe: 26716.7 eV, $Pb_{0.09}Cd_{0.91}Se$: 26716.4 eV, and $Pb_{0.4}Cd_{0.6}Se$: 26717.6 eV (note: the step size of 1 eV used in the XANES measurements), in good agreement with the XPS results. The prominent scattering pathway at 2.2 Å in the extended X-ray absorption fine structure (EXAFS) spectra of the Cd K-edge is attributed to the interatomic scattering pathway of Cd-Se (Supplementary Fig. 8). The EXAFS data fitting results in Supplementary Table 4 show a Cd-Se distance of 2.571 to 2.606 Å with a coordination number of 2.8 to 3.8 for these three samples (bulk = 4), which is close to the theoretical coordination number for ~3 nm CdSe nanoparticles (the coordination number expected for a 25-Å spherical particle is 2.9-3.8, depending on the exact size and origin of the sphere with respect to the lattice).⁵⁹ These results suggest a similar chemical environment for Cd in $Pb_xCd_{1-x}Se$ QD gels. However, because of the X-ray absorption interference between the Se K-edge (12657.8 eV) and the Pb L-edge (13035.2 eV), the XAS signal of the Pb L_{III} edge was too noisy at the edge region for EXAFS analysis to obtain the coordination environment of Pb in the gel samples (Supplementary Fig. 9a, d)."

10. I found the term 'isolated Pb site' confusing, thus, I would recommend revisiting the terms and use e.g. ionic sites and segregated sites (or something more distinguishable) to fully separate the atomically dispersed Pb from the Pb in PbSe matrix. This would improve the readability (especially in the DFT and mechanism parts) and understand the main message: ionic sites are superior compared to segregated PbSe sites from sensing performance point of view.

*Author reply: An excellent point. We have adjusted the term "isolated Pb site" to "**atomically dispersed Pb ionic site**" in the main text.*

11. The novelty is remarkable, the conclusion is supported by the data, and after revision, the paper will definitely have the community's attention. The method is well-described in the experimental section for reproducibility. After addressing the major concerns, the sensing performance warrants the paper to be published in Nature Communications.

Author reply: We thank the reviewer for the positive opinion about our work.

Reviewer 4

Comments:

I would recommend authors to clarify the following issues:

1. Authors claim that “Short-term exposure to a high concentration of NO₂ irritates the human respiratory system, causing respiratory distress symptoms such as coughing, wheezing, and difficulty breathing.” Sensing strategy authors proposed implies, however, exploitation of toxic heavy metals (Cd and Pb) that can adversely affect the human health in their own way. In many countries, the use of these hazardous materials in electrical and electronic devices is undesirable. Could authors justify somehow utilization of Cd- and Pb-containing materials for environmental sensing?

Author reply: The reviewer made a good point. The RoHS (the Restriction of the use of certain hazardous substances) sets the International Standards for lead and cadmium in electrical and electronic devices (maximum concentration by weight) as 0.1 % (1000 ppm) and 0.01% (100 ppm), respectively.¹⁰ The Pb and Cd content in our Pb_xCd_{1-x}Se QD gel sensors fulfills the RoHS standards. Taking a Pb_{0.09}Cd_{0.91}Se gel sensor as an example, the Pb and Cd weight in the sensor is approximately ~9.3 µg and ~51 µg, respectively. The weight of our sensor device is ~100 g. Therefore, the Pb and Cd concentrations of Pb_{0.09}Cd_{0.91}Se gel sensor by weight are estimated to be 0.1 ppm and 0.5 ppm, several orders of magnitude lower than the RoHS standards.

We added the following

Page 8: “It is worth noting that the Pb and Cd contents by weight in this home-built NO₂ sensing device are 0.1 ppm and 0.5 ppm, respectively; both are several orders of magnitude lower than the International Standards for Pb (0.1 % or 1000 ppm) and Cd (0.01% or 100 ppm) in electrical and electronic devices set by the RoHS (Restriction of the use of certain hazardous substances).⁶⁵”

2. Alumina was used as a sensor substrate. It would be good to add several sentences explaining why did authors choose this substrate? What is a strength of gel adhesion to the surface of the alumina substrate? Is there any interaction between the substrate and Pb_xCd_{1-x}Se QD gel? Could this interaction potentially affect the sensing performance?

*Author reply: We thank the reviewer for the great comment. Actually, both alumina (for lab sensing tests) and silicon (for wireless portable device) substrates were used in this work. Alumina and silicon are the most commonly used and commercially available sensor substrate materials due to their high temperature resistance, excellent insulation, and cost-effectiveness. The Pb_xCd_{1-x}Se QD gel sensors with different substrates exhibit very comparable sensing performance, so we do not think the substrates interact with the Pb_xCd_{1-x}Se QD gels and affect the sensing performance. The excellent long-term stability of the QD gel sensor (Figure 4e) under a continuous gas flow for **75 hours** indicates that the adhesion strength between Pb_xCd_{1-x}Se QD gels and substrates should be high.*

We added the following to the main text.

Page 7: “For regular sensing tests, commercial alumina substrates were used because of their cost-effectiveness. For the homebuilt wireless portable sensing device tests, the silicon-based substrates were fabricated using photolithography to be compatible with the device (Supplementary Fig. 16-17).”

Page 8: “... suggesting the strong gel adhesion to the substrate and high stability of the gel sensors.”

3. On Page 8, authors introduce palladium sites. Probably this is a typo. Otherwise, authors need to explain a role of Pd in more details.

line 23 “For all bimetallic surfaces, their Cd sites adjacent to Pd show stronger adsorption energies and larger”

line 37 “The above findings show that (1) the Cd sites adjacent to Pd cations in $Pb_xCd_{1-x}Se$ surface lead to”.

Author reply: We thank the reviewer for pointing out those typos, which have now been corrected from “Pd” to “Pb” in the main text.

4. Page 6, line 26-30. Authors wrote “ $Pb_xCd_{1-x}Se$ gel sensors were prepared by drop-casting 10 μL of wet gel onto a sensor substrate patterned with an interdigitated electrode, followed by drying in air. The dried gel film exhibits a highly porous surface morphology (Supplementary Fig. 14) and a thickness of $\sim 3.1 \mu m$ (Supplementary Fig. 15).”

Is there any special reason why did authors prepare the gel film of $\sim 3.1 \mu m$ thickness? Is this thickness optimal? Will nm-thick gel films do have other sensing performance?

*Author reply: Excellent point. Following the reviewer's suggestion, we studied the effect of gel film thickness on the sensing performance. We prepared gel sensors with different thicknesses by varying the amount of wet gel for drop-casting. Taking $Pb_{0.09}Cd_{0.91}Se$ gel as an example, we drop-casted 2 μL , 10 μL , and 50 μL wet gel onto the sensor substrates to prepare 0.7 μm , 3.1 μm and 18.5 μm -thick films (**Figure R7a-b**). The sensing results show that the thickness does affect the sensor response (S), response time (t_{res}) and recovery time (t_{rec}) (**Figure R7d and Table R7**). As the film thickness decreases, S increases, and t_{res} and t_{rec} decrease, affording higher sensing performance. However, the slight improvement ($\sim 5\%$) in S , t_{res} , and t_{rec} when the film thickness is reduced from 3.1 μm to 0.7 μm comes with a high price of a five times higher sensor resistance ($\sim 30 M\Omega$ vs. 6 $M\Omega$), approaching to the detection limit of our resistance-measuring instrument. Such resistance increase causes more problems for studying gels with higher electrical resistivity than $Pb_{0.09}Cd_{0.91}Se$ gel, for example, $CdSe$ and $Pb_{0.003}Cd_{0.997}Se$ gels, because the resistance of their 0.7 μm -thick films is over the instrument detection limit of 120 $M\Omega$, making the systematic study of the effect of Pb content impossible. As a result, 3.1 μm is currently the optimal thickness for our study.*

We added **Figure R7** as **Supplementary Figure 19**, and **Table R7** as **Supplementary Table 5** to the supplementary information. The following sentences are added to the main text.

Page 7: “The gel film thickness can be varied by controlling the amount of wet gel deposited onto the substrate. Although thinner films are expected to afford higher sensing performance, including higher response and faster response and recovery, the reduced film thickness dramatically increases the sensor resistance, imposing a technical challenge in measuring resistance (Supplementary Fig. 19 and Supplementary Table 5). As a result, we performed all the sensing tests at the film thickness of 3.1 μm .”

Figure R7. a-c, FE-SEM cross-sectional images of $\text{Pb}_{0.09}\text{Cd}_{0.91}\text{Se}$ gel sensors prepared by drop-casting 2 μL , 10 μL and 50 μL wet gel onto the sensor substrate, respectively. d, Response-recovery curves of $\text{Pb}_{0.09}\text{Cd}_{0.91}\text{Se}$ gel sensors with a thickness of 0.7 μm , 3.1 μm , and 18.5 μm at 1.32 ppm NO_2 at room temperature.

Table R7. Comparison of sensing performance for $\text{Pb}_{0.09}\text{Cd}_{0.91}\text{Se}$ gel sensors with different thickness towards 1.32 ppm NO_2 at room temperature.

	Sensor response (S , %)	Response time (t_{res} , s)	Recovery time (t_{rec} , s)	Base resistance ($M\Omega$)
0.7 μm	75.5	26	61	31.7
3.1 μm	72.8	28	66	5.8
18.5 μm	66.7	35	82	1.0

5. As the isolated atomic sites in hexagonal matrix authors considered Pb^{2+} divalent ions. From the theoretical point of view, the presence of charged ions will cause the formation of surface dipole. Did authors apply the dipole correction to compute a total energy of certain system? Should Pb on the surface of a CdSe be treated as a charged ion or as a neutral atom? In other words, the system containing isolated ions was neutral or charged?

Author reply: We appreciated the reviewer's comment. The overall model is neutral, with Cd and Pb at a valence of +2 and Se at -2. To avoid confusion, we replaced Pb^{2+} in the main text with Pb ionic sites. Following the reviewer's suggestion, we tested if the dipole correction affects the calculated energies. The calculated total energy and NO_2 adsorption energy for the D2 geometry in Figure 5b show no noticeable difference with and without dipole correction (**Table R8**), suggesting dipole correction is not required.

Table R8. Dipole correction for D2's total energy and NO_2 adsorption energy

Items	Energy / eV	
	without dipole correction	with dipole correction
D2 adsorption geometry	-298.43	-298.42
slab of D2 adsorption geometry	-279.56	-279.55
NO_2 adsorption energy of D2	-0.80	-0.80

References

1. D. Kim, B. Yoo. *Sensors and Actuators B* 2011, 160, 1168-1173.
2. P. K. Jain, L. Amirav, S. Aloni, A.P. Alivisatos. *J. Am. Chem. Soc.* 2010, 132, 9997– 9999.
3. H. Li, M. Zanella, A. Genovese, M. Povia, A. Falqui, C. Giannini, L. Manna. *Nano Lett.* 2011, 11, 4964– 4970.
4. J.M. Luther, H. Zheng, B. Sadtler, A.P. Alivisatos. *J. Am. Chem. Soc.* 2009, 131, 16851– 16857.
5. I.R. Pala, S.L. Brock. *ZnS nanoparticle gels for remediation of Pb^{2+} and Hg^{2+} polluted water.* *ACS Appl. Mater. Interfaces* 2012, 4, 2160-2167.
6. Q. Yao, I.U. Arachchige, S.L. Brock. *Expanding the Repertoire of Chalcogenide Nanocrystal Networks: Ag_2Se Gels and Aerogels by Cation Exchange Reactions.* *J. Am. Chem. Soc.* 2009, 131, 2800-2801.
7. F. Helfferich. *Ion Exchange*; McGraw-Hill Book Company: New York, 1962; p 2.
8. K.L. Silva, L. Silmi, S.L. Brock. *J. Chem. Phys.* 2019, 151, 234715.
9. J. L. Davis, A. M. Chalifoux and S. L. Brock, *Langmuir*, 2017, **33**, 9434-9443.
10. European Commission, *Restriction of Hazardous Substances in Electrical and Electronic Equipment.* https://ec.europa.eu/environment/topics/waste-and-recycling/rohs-directive_en (accessed by 21 July 2011).

REVIEWER COMMENTS

Reviewer #1 (Remarks to the Author):

The authors have addressed my concerns and I believe the manuscript is now suitable for publication.

Reviewer #2 (Remarks to the Author):

I found that the authors sincerely and completely responded to the reviewers' comments and think that the manuscript is good enough to be published in Nature Communication.

Reviewer #3 (Remarks to the Author):

The Authors have sufficiently addressed the emerged concerns and points. Although the response on the gelation of cubic/hexagonal CdSe is not fully satisfactory, the reason of not extending and optimizing the gelation on cubic CdSe QDs (to prepare higher quality, cubic PbSe) in the present work is understandable and indeed points beyond the scope of this study. Therefore, I recommend the paper for publication in its revised form.

Reviewer #4 (Remarks to the Author):

Authors addressed all reviewers' questions, made required changes and significantly improved the quality of the manuscript. Reviewer found the revised version of the paper suitable for publication in Nature Communications.

Response to Reviewers' Comments

Ms. ID: NCOMMS-21-15323

Title: Atomically dispersed Pb ionic sites in PbCdSe quantum dot gels enhance room-temperature NO₂ sensing

Author response in italics

Reviewer 1

Comments:

The authors have addressed my concerns and I believe the manuscript is now suitable for publication.

Author reply: We thank the reviewer for the positive opinion about our work.

Reviewer 2

Comments:

I found that the authors sincerely and completely responded to the reviewers' comments and think that the manuscript is good enough to be published in Nature Communication.

Author reply: We thank the reviewer for the positive opinion about our work.

Reviewer 3

Comments:

The Authors have sufficiently addressed the emerged concerns and points. Although the response on the gelation of cubic/hexagonal CdSe is not fully satisfactory, the reason of not extending and optimizing the gelation on cubic CdSe QDs (to prepare higher quality, cubic PbSe) in the present work is understandable and indeed points beyond the scope of this study. Therefore, I recommend the paper for publication in its revised form.

Author reply: We thank the reviewer for the positive opinion about our work.

Reviewer 4

Comments:

Authors addressed all reviewers' questions, made required changes and significantly improved the quality of the manuscript. Reviewer found the revised version of the paper suitable for publication in Nature Communications.

Author reply: We thank the reviewer for the positive opinion about our work.